# A Lifetime of Catalytic Micro-/Nanomotors

**DOI:** 10.3390/nano15010013

**Published:** 2024-12-26

**Authors:** Tao He, Yonghui Yang, Xuebo Chen

**Affiliations:** School of Electronic and Information Engineering, University of Science and Technology Liaoning, Anshan 114051, China; ht18191863591@163.com (T.H.); 319953100072@ustl.edu.cn (Y.Y.)

**Keywords:** catalytic micro-/nanomotors, preparation mode, driving mechanism, control strategy, application

## Abstract

Microscopic and nanoscopic motors, often referred to as micro-/nanomotors, are autonomous devices capable of converting chemical energy from their surroundings into mechanical motion or forces necessary for propulsion. These devices draw inspiration from natural biomolecular motor proteins, and in recent years, synthetic micro-/nanomotors have attracted significant attention. Among these, catalytic micro-/nanomotors have emerged as a prominent area of research. Despite considerable progress in their design and functionality, several obstacles remain, especially regarding the development of biocompatible materials and fuels, the integration of intelligent control systems, and the translation of these motors into practical applications. Thus, a comprehensive understanding of the current advancements in catalytic micro-/nanomotors is critical. This review aims to provide an in-depth overview of their fabrication techniques, propulsion mechanisms, key influencing factors, control methodologies, and potential applications. Furthermore, we examine their physical and hydrodynamic properties in fluidic environments to optimize propulsion efficiency. Lastly, we evaluate their biosafety and biocompatibility to facilitate their use in biological systems. The review also addresses key challenges and proposes potential solutions to advance their practical deployment.

## 1. Introduction

Movement is a fundamental aspect of life for organisms in both macroscopic and microscopic environments, as it is crucial for sustaining various biological processes. Through the course of evolutionary development, nature has crafted highly efficient and sophisticated biomolecular motor proteins that facilitate a wide range of cellular and biological functions [1,2,3]. For instance, bacteria employ rotating flagellar nanomotors to propel themselves [1]. Similarly, motor proteins such as kinesin, myosin, and dynein enable directed movement along microtubules and actin filaments by utilizing the energy released from ATP hydrolysis [1]. Additionally, ATPase, a specialized biomolecular engine, generates ATP, which serves as the biofuel that powers cellular activities and supports essential life processes [3].

Catalytic micro-/nanomotors (MNMs) are nanoscale systems capable of converting chemical energy into mechanical motion or force. As discussed earlier, through the process of biological evolution, nature has equipped biomolecular motor proteins with the ability to capture and utilize environmental energy, enabling autonomous movement within living organisms. In the past decade, significant progress has been made in the development of synthetic MNMs, drawing inspiration from these natural biomolecular motors. The research efforts of the team led by Sen and Mallouk have been particularly influential in this field. Currently, much of the research is concentrated on the design of catalytic MNMs that can efficiently and rapidly convert chemical energy into self-directed motion [4,5,6].

Over the past decade, the field of synthetic self-propelled catalytic MNMs has experienced considerable progress, marked by several important advancements and contributions [4,5,6,7,8,9]. While the future prospects of this research area are promising, numerous challenges persist. For instance, the design, synthesis, and characterization of functional MNMs necessitate the development of innovative methods and conceptual approaches to achieve the desired performance. Furthermore, the fabrication of MNMs with specific functional components and the precise, intelligent control of their motion characteristics remain substantial hurdles. Another ongoing challenge is the identification of suitable applications, including biocompatible fuels. To improve the control over the motion of catalytic MNMs and to make their industrial-scale implementation feasible, it is essential to gain a deeper understanding of the physicochemical principles that govern their behavior.

Catalytic MNMs have a broad range of potential applications spanning various fields, such as load transportation, wastewater remediation, chemical sensing, and biomedicine [5,7,8,9,10,11,12,13,14,15,16,17,18,19,20,21,22,23,24,25,26,27,28,29,30,31,32,33,34]. As the development of more advanced MNMs progresses, it is expected to drive further innovations in these interconnected areas. Nevertheless, the design and propulsion of MNMs continue to present substantial challenges in the realm of nanotechnology. To overcome these obstacles and improve performance, it is essential to thoroughly understand the current research landscape surrounding MNMs. This paper seeks to explore and analyze the fabrication techniques, propulsion mechanisms, control methodologies, and possible applications of self-propelled catalytic MNMs (Figure 1).

## 2. Preparation Method

Artificial MNMs are crucial components in the development of miniaturized devices. A major obstacle in this field, however, is the ability to synthesize MNMs that are both high-quality and reliable. Since each unique MNM design demands a specific synthesis technique and corresponding propulsion mechanism, it is important to select the appropriate fabrication approach based on the MNMs’ intended function. With the rapid progress in nanotechnology, a variety of fabrication methods for MNMs have emerged. This section offers an overview of the current synthesis techniques for MNMs. Key factors such as shape, material composition, distribution, and the integration of functionalized components must be carefully considered during the fabrication process. By examining the advancements made in MNM synthesis over the past decade, we aim to identify both the challenges faced and the opportunities arising in this area, while also providing insights into future directions for developing new and innovative synthesis methods.

### 2.1. Electrochemical Deposition

Electrochemical deposition, also known as electrodeposition, is a process that involves the growth of materials through the application of an external electric current. One of the key advantages of this method is its versatility in creating a diverse range of materials, including metals and polymers, in complex three-dimensional forms. This ability has made electrodeposition a popular and widely utilized technique in nanotechnology. Additionally, the process is cost-effective, requiring minimal specialized equipment, and can be carried out in relatively simple laboratory conditions. Consequently, electrodeposition allows for the production of various micro- and nanostructures, such as nanowires, nanorods, and microtubes, with a broad spectrum of sizes.

Template-assisted electrochemical deposition involves the use of membrane templates with perforations, which guide the growth of specific nanowires and microtubes made from a variety of materials [35]. Each pore in the membrane functions as a miniature reactor, where the desired micro-/nanostructures are formed. Track-etched polycarbonate (PC) membranes and porous alumina (AAO) membranes are the most commonly employed templates for fabricating these micro-/nanostructures. The physical properties and chemical composition of the pore walls play a critical role in determining whether the resulting MNMs are solid or hollow. This method, which relies on membrane templates for electrodeposition, provides an efficient and cost-effective approach to synthesizing nanowires, nanorods, and microtubes.

In membrane template-assisted electrodeposition, a thin Au/Ag layer is initially deposited onto one side of the membrane using physical vapor deposition (PVD), which serves as the working electrode for the subsequent process. The membrane is then placed into a Teflon plating cell, where a flat Al foil is positioned near the metal layer to act as the conductive contact for electrodeposition. Typically, a sacrificial Ag/Cu layer is first deposited, followed by the targeted metal layers in sequence. After deposition, the Ag/Au substrate and sacrificial layers are removed through chemical etching or mechanical polishing. The Al_2_O_3_ membrane is then dissolved using a NaOH solution, followed by several rinsing and centrifugation steps. This series of procedures ultimately releases the nanowires or nanotubes from the template.

The research teams led by Wang and Pumera have introduced an innovative approach that integrates electrodeposition—a widely utilized technique for producing nanowire-based MNMs—with bubble-propelled tubular micro-/nanojets (Figure 2) [36,37,38,39,40,41]. This method allows for the fabrication of cylindrical or conical MNMs, with the specific shape of the MNMs being influenced by the type and geometry of the porous template used. In addition to the metals previously mentioned, the combination of polymers such as polyethylenedioxythiophene (PEDOT), polyaniline (PANI), and polypyrrole (PPy) with Pt also yields catalytic MNMs. Moreover, the walls of these MNMs can be functionalized with molecularly imprinted polymers (MIPs), which provide selective recognition cavities for the targeted separation of biomolecules.

To address the challenges posed by complex fluid environments on the kinematic behavior of catalytic MNMs, researchers have developed non-regularly shaped catalytic MNMs. For instance, Liu et al. fabricated Pd nanosprings through electrochemical deposition, utilizing an anodic Al_2_O_3_ membrane supported by nanochannels [42]. The hydroxyl-terminated surfaces of these nanochannels can selectively absorb H^+^, forming a compact layer under specific acid–base conditions. When exposed to an effective potential and an electroplating solution containing PdCl_2_, CuCl_2_, and HCl, the hydroxyl-terminated surfaces of the Al_2_O_3_ nanochannels, combined with localized hydrogen precipitation, promote the deposition of Pd atoms at the outer edges of the nanochannels. Due to helical dislocation, Pd is naturally wound around Cu nanorods, and after selective removal of Cu, Pd nanosprings are formed. By modifying the nanopore diameter and ion concentration in the plating solution, template-assisted electrodeposition allows for control over the diameter and length of helical MNMs. Additionally, magnetic materials can be incorporated onto the surface of these helical structures to enable precise magnetic navigation [43].

In electrochemical deposition, the template used can be a porous membrane, with its shape chosen based on the desired structure to be created. To streamline the electrodeposition process, Manesh et al. developed a simplified template-assisted method to fabricate catalytic conical MNMs [44]. This technique involved sequentially depositing Pt and Au onto etched Ag wires, which were subsequently diced and dissolved. This approach allows for precise control over the properties of the catalytic MNMs, thereby improving their performance. The motion of these catalytic MNMs is driven by an internally generated O_2_ microbubble, which creates a recoil effect, leading to salt-independent movement. This mechanism overcomes the limitations of traditional catalytic nanowires, where ionic strength typically restricts their motion. However, it is important to note that this method is not ideal for batch production, and the velocity of the resulting MNMs is relatively modest.

### 2.2. Physical Vapor Deposition

PVD is a process used to deposit thin layers of material onto a substrate. In this process, the material is first vaporized from a solid target, either by exposure to a gaseous plasma or under high-temperature vacuum conditions. The vaporized material is then directed toward the substrate surface in a vacuum/partial vacuum environment. Upon reaching the substrate, the vapor condenses and forms a thin film. The two most commonly used PVD techniques are electron beam evaporation and sputtering. In electron beam evaporation, a focused electron beam is used to directly vaporize the target material, causing its atoms to transition into the gas phase. Conversely, sputtering involves bombarding the target material with ionized gas, usually Ar, to generate vapor. The vapor produced by both techniques eventually condenses onto the substrate surface to form the desired thin film.

The use of PVD for the fabrication of catalytic MNMs has emerged as a highly effective technique. In comparison to template-assisted electrochemical deposition, PVD offers several notable advantages, such as the ability to deposit a diverse range of materials, reduced preparation steps, simplified handling processes, and the flexibility to create MNMs with more intricate shapes. A critical factor in PVD is the deposition angle, which significantly influences whether the process follows traditional growth patterns or dynamic shadowing growth (DAG), also referred to as glancing angle deposition (GLAD). In conventional PVD, the substrate is aligned parallel to the target material, allowing the vaporized metal to condense onto the substrate in a nearly vertical orientation. In contrast, DAG involves directing the metal vapor toward the substrate at a predetermined tilt angle, leading to a different deposition morphology.

Posner and colleagues developed bimetallic spherical MNMs that operate based on electrophoretic motion (Figure 3A) [45]. To prepare the microspheres, they first underwent a sputtering deposition process, where half of their surface was coated with Au. Afterward, the spheres were re-suspended in water and repeatedly coated with Au in random directions. This process was performed seven to eight times, ensuring that the entire surface of the microspheres became uniformly coated with Au. The final step involved depositing Pt on the other half of the spheres. The resulting bimetallic spherical MNMs exhibited motion speeds similar to those of nanowires. Beyond spherical templates, PVD can also be utilized to fabricate other Janus MNMs, as well as various other geometrically diverse MNMs. In another study, Valadares and collaborators explored the fabrication of a catalytic dimer made up of a Pt hemisphere and SiO_2_ spheres. The preparation began with the sputtering of a Cr/Pt bilayer onto the sphere, followed by an annealing step. During this process, the metallic half-shell formed a Pt particle that was integrated with the SiO_2_ sphere (Figure 3B) [46].

The combination of substrate rotation and the self-shadowing effect during vapor deposition provides a more efficient means of fabricating Janus MNMs with intricate geometries, as illustrated by the DAG technique. Zhao and colleagues explored the creation of asymmetric catalytic MNMs with Pt/Au coatings using the GLAD method. To achieve the desired asymmetric bimetallic deposition, the substrate, coated with SiO_2_ microbeads, was rotated to a polar angle after depositing a Ti and Au adhesive layer. This rotation allowed for partial exposure of the Au layer during subsequent Pt coating (Figure 3C) [47]. The propulsion of these MNMs can be fine-tuned by altering the extent of the exposed Au surface area. In another study, Lee and coworkers synthesized Pt/Au Janus MNMs with a diameter of approximately 30 nm using DAG. During the fabrication process, Au was deposited onto a Pt nanoparticle array, which was prepared using block copolymer micelle lithography, all while the substrate was rapidly rotated [48]. Both types of bimetallic Janus MNMs utilize self-electrophoresis for their propulsion mechanism.
Figure 3MNMs prepared by physical vapor deposition. (**A**,**B**) Preparation of MNMs by conventional physical vapor deposition. (**A**) Schematic of fabrication of bimetallic Janus micromotors by conventional physical vapor deposition. Reproduced from Ref. [45]. Copyright 2010, the American Chemical Society. (**B**) Formation of sphere dimers via thermal annealing. Reproduced from Ref. [46]. Copyright 2009, Wiley-VCH. (**C**–**E**) Preparation of MNMs by glancing angle deposition. (**C**) Preparation of asymmetric Pt/Au-coated catalytic micromotors by GLAD. Reproduced from Ref. [47]. Copyright 2010, the American Institute of Physics. (**D**) Fabrication procedure of L-shaped Si/Pt nanorod motors by GLAD. Reproduced from Ref. [49]. Copyright 2007, the American Chemical Society. (**E**) Synthesis of catalytic micromotor consisting of a spherical silica colloid with a TiO_2_ arm coated asymmetrically with Pt. Reproduced from Ref. [50]. Copyright 2009, Wiley-VCH.
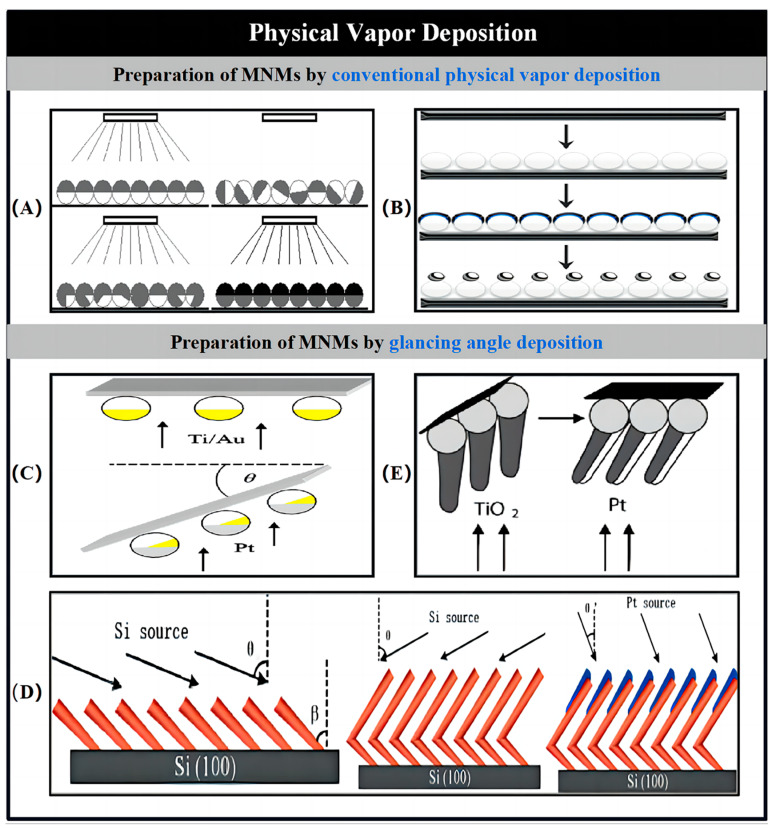



He and his research team successfully demonstrated the fabrication of rotating Si/Pt and Si/Ag nanorods, as well as L-shaped Si/Pt nanorods. The Si nanorod structure was initially formed using the DAG technique. This was followed by asymmetric deposition of a Pt/Ag layer, which was applied to one side of the nanorod skeleton using geometric shading effects. For the creation of L-shaped nanorods, the substrate was rapidly rotated azimuthally during the middle phase of the tilt angle deposition process (Figure 3D) [49]. By carefully controlling the deposition angle and substrate rotation, more complex shapes, such as rolling Si/Ag springs, can be fabricated. In another study, Gibbs and Zhao reported the rotational propulsion of MNMs consisting of SiO_2_ microbeads and TiO_2_ arms with asymmetric Pt deposition. The arms were deposited at an inclined angle over the closely spaced microbeads, resulting in Pt coating being applied exclusively to one side of the arms, which is an essential method for achieving asymmetric catalyst positioning (Figure 3E) [50]. The combination of substrate rotation and controlled gas-phase direction tilting allows for a more extensive coverage of the sphere template than what conventional gas-phase deposition methods can achieve. Following this, Pt-Ag-Au shell-like MNMs were produced through DAG and wet chemical etching. The MNMs’ surface features a small aperture, which plays a critical role in facilitating bubble-driven propulsion [51].

### 2.3. Rolled-Up Technology

By incorporating a precisely engineered stress gradient and applying a coated film, the film can be made to curl into a predetermined shape upon separation from the substrate. The controlled application of strain to the deposited layer promotes the formation of microtubes through a managed rolling mechanism. To create uniformly stressed nanofilms, these films are coated onto a patterned photoresist sacrificial layer, which is fabricated through photolithography. Afterward, the sacrificial layer is removed by etching with acetone. The use of DAG technology ensures that microtubes can be accurately positioned and integrated on a single chip. By adjusting key parameters such as deposition rate, substrate temperature, and the evolution of strain during the coating process, the necessary stress gradient for rolling the film into microtubes can be effectively controlled.

Once the sacrificial layer is dissolved, the coated nanofilm detaches from the substrate, resulting in the formation of microtubes (Figure 4A) [52,53]. To ensure that the rolled nanofilms do not collapse, it is crucial to dry the microtubes at a temperature close to the critical drying point. By adjusting factors such as the inherent stress and the thickness of the nanofilm, microtubes with opening diameters ranging from 1 to 30 μm can be successfully fabricated. Typically, these microtubes have lengths on the order of several tens of micrometers. Applying a catalyst like Pt to the top layer of the nanofilm facilitates the creation of a catalytic inner wall once the film is rolled. Additionally, the orientation of the folding process is influenced by the varying etching rates along the crystal axes and the crystal structure of the sacrificial layer, which determine how the film separates.

Due to the high costs and complexity of the rolled-up fabrication method, significant research has been focused on simplifying the process and reducing its expenses. As illustrated in Figure 4B, a microtube with a graphene oxide (GO) outer layer was created by applying a thin metal coating onto GO nanosheets [54]. The inherent strain in the material, along with the weak interlayer bonds of GO, facilitates the spontaneous formation of microrolls with GO on the outside and Pt on the inside when subjected to ultrasound. Additionally, the size of these microrolls can be controlled by adjusting the thickness of the metal coating. In a similar vein, tubular MNMs were fabricated using fruit cells as inexpensive scaffolds for the metal layer. This bio-based microengine demonstrated highly efficient bubble propulsion in H_2_O_2_ solutions. Zhao and collaborators also developed Pt microtubes by either dissolving a poly(methyl methacrylate) (PMMA) sacrificial layer under the sputtered Pt coating or directly depositing Pt onto a glass wafer, followed by lift-off in the presence of H_2_O_2_. This approach produced relatively uniform microtubes, utilizing a transmission electron microscopy (TEM) grid template [54]. While these affordable and simplified techniques hold promise for large-scale production, they still face challenges in achieving precise control over the morphology and size of the microtubes. Consequently, future work should focus on improving the roll-up process while maintaining its cost-effective nature.

Magdanz and collaborators developed microjets made from a thermo-responsive polymer that undergoes reversible folding and unfolding in response to temperature fluctuations (Figure 4C) [55]. As the temperature drops, the Pt/polymer composite cools, which induces the film to automatically fold into microtubes with a diameter of around 30 μm. These microtubes feature an inner Pt layer that functions as a catalyst. Upon increasing the temperature, the microtubes return to their initial, unfolded form, allowing for a cycle of rolling and unrolling. By varying the temperature of the surrounding solution, the motion of these microtubes can be controlled, enabling either activation or cessation of movement. The microtubes created through this rolled-up technique maintain diameters in the micrometer range.

Li and colleagues successfully achieved a reduction in the diameter of microtubes fabricated using the roll-up technique, bringing it down to the submicrometer scale. This was accomplished by utilizing the surface tension of nanodroplets combined with the strain relaxation properties inherent in nanofilms. As depicted in Figure 4D, a Pt layer was first deposited on a PMMA sacrificial layer supported by pre-stressed bilayer structures made of SiO_2_/TiO_2_ or Si/Cr [56]. During the deposition process, rapid thermal processing (RTP) was applied to the Pt layer, which caused it to stretch and form isolated islands. The generated nanodroplets played a crucial role in providing the surface tension needed for the rolling mechanism. Following the removal of the PMMA sacrificial layer, the nanofilm detached and microtubes were formed. These microtubes exhibited enhanced propulsion velocities in H_2_O_2_ solutions compared to those with uniform Pt surfaces.

The strain within thin material layers can be harnessed to transform a straight band structure into a magnetic helical configuration through the rolling process [57]. For example, Zhang and colleagues developed helical MNMs using a conventional thin film deposition technique [58]. Initially, a metal layer was deposited and then shaped into straight ribbons through reactive ion etching. Chemical vapor deposition was employed to form a magnetic head for magnetic control. The helical structure emerged due to the material’s self-rolling behavior, which occurs when the ribbons are detached from the substrate, driven by internal pressure within the material. The properties of these helical MNMs can be fine-tuned by adjusting factors such as film thickness, ribbon width, and the orientation of the ribbons relative to the crystalline structure of the metal. To explore the kinematic properties of these structures, Li and colleagues examined how the size of the magnetic head influences the speed of the helical materials [59]. At lower frequencies, the helical structures with smaller magnetic heads exhibited faster movement due to reduced viscous drag in the fluid. However, it was also observed that the larger magnetic heads, which contain more Ni, experience a stronger magnetic moment, resulting in a higher maximum velocity.

Helical MNMs, produced through thin film deposition, can also be propelled by electro-osmotic forces induced by electric fields, in addition to the use of rotating magnetic fields [60]. Electro-osmotic actuation operates through the interaction between the surface of the helical structure and the surrounding fluid solution. When an external electric field is applied, it causes the movement of the Stern layer on the helical structure’s surface, which in turn generates hydrodynamic pressure. This pressure acts on the surface of the helical material, driving it in the opposite direction. The electro-osmotic actuation mechanism improves the traveling speed and maneuverability of the helical structure, offering advantages over the use of rotating magnetic fields.

### 2.4. Advanced Assembling

The previously discussed methods, such as template-assisted electrochemical deposition, PVD, and rolled-up nanotechnology, have shown effectiveness in synthesizing artificial MNMs. However, as the complexity of the structures increases, researchers have been compelled to develop more advanced assembly techniques. One of the primary challenges in this area is the construction of devices made up of multiple independent microcomponents. The assembly process, which involves integrating miniaturized components into intended structures, plays a crucial role in the fabrication of MNMs. These techniques not only rely on the self-assembly of materials to create specific devices but also allow for the incorporation of targeted elements into the MNMs by embedding them within the material matrix.

Layer-by-layer (LbL) self-assembly is a flexible and effective technique in nanofabrication, enabling the creation of multilayered films through the sequential deposition of materials with opposite charges. This method is both straightforward and economical, allowing for the encapsulation of a wide variety of substances, such as inorganic compounds, colloids, macromolecules, and organic molecules. The LbL process is highly adaptable, suitable for a range of solvent-accessible surfaces, and can accommodate different templates during fabrication. For example, incorporating Pt nanoparticles into the multilayer structure imparts motility in H_2_O_2_ solutions. LbL assembly is primarily driven by electrostatic interactions between the oppositely charged materials, and it has become a widely adopted approach for the synthesis of diverse multilayered materials due to its efficiency, versatility, and ease of implementation.

He’s group was at the forefront of developing autonomous Janus MNMs, which were asymmetrically coated with Pt nanoparticles, through an innovative combination of colloid template-assisted LbL assembly and microcontact printing techniques (Figure 5A) [61]. In this approach, SiO_2_ particles were used as templates and were selectively dispersed in solutions of positively charged polyallylamine hydrochloride (PAH) and negatively charged polystyrene sulfonate (PSS), which facilitated the formation of polyelectrolyte bilayers. After five bilayers were applied, the particles were transferred to a glass wafer, where they were arranged into a monolayer. This monolayer was then printed using a PDMS stamp, which was loaded with a suspension of dendritic Pt nanoparticles. Following the removal of the template using HF, hollow Janus MNMs containing Pt nanoparticles on one side were obtained. In a similar study, Wilson and his team demonstrated the successful encapsulation of Pt nanoparticles within nanocavities of polymer stomatocytes (Figure 5B) [62]. This process involved the controlled transformation of spherical polymer vesicles into stomatocyte-like structures, where Pt nanoparticles were introduced into the solvent-swollen vesicles and captured. Moreover, the team achieved directional propulsion for the stomatocytes by carefully controlling the inlet structure, allowing H_2_O_2_ to interact with the encapsulated Pt nanoparticles and generate propulsion.

### 2.5. Summary

To perform intricate tasks, MNMs require the integration of multiple functional modules to ensure their effective operation. These modules typically include a structural framework, an actuator for autonomous movement, and a surface material that interacts with the external environment via intrinsic properties or surface functional groups [63]. The process of incorporating these functional modules into MNMs follows two main approaches: the top-down and bottom-up strategies. The top-down approach begins by forming a macroscopic structure, which is then miniaturized to the microscale through methods such as PVD and rolled-up nanotechnology [64,65]. In contrast, the bottom-up strategy constructs the structure starting at the atomic or molecular level, gradually assembling it into the desired microscale form through techniques like electrodeposition and self-assembly [65]. Although the top-down approach often entails more complexity and extended preparation times, it provides precise control over design and enhanced performance. Meanwhile, the bottom-up method offers advantages such as simplicity, scalability, and suitability for large-scale industrial production. Both strategies have distinct benefits (Table 1). Future advancements should explore how to effectively combine these approaches to develop MNM systems that are high-performing, stable, and adaptable to complex environments.

## 3. Propulsion Mechanism

Whitesides and his research team were the first to demonstrate the movement of a millimeter-sized object, which consisted of a Pt-coated porous glass filter affixed to a thin PDMS plate using a stainless steel pin [66]. Upon immersion in an H_2_O_2_ solution, the Pt catalyst triggers the decomposition of H_2_O_2_, producing oxygen bubbles. These bubbles detach from the catalyst’s surface, generating a recoil force that drives the object forward. This discovery laid the foundation for the field of chemically powered motion. At the micro- and nanoscale, autonomous propulsion is exclusively achievable with asymmetric particles. As a result, a variety of MNMs have been designed by altering their symmetry, either in shape or material composition. Several mechanisms have been proposed to explain the propulsion behavior, which depend on the specific shape and material properties of the MNMs [29,30,31,32,33,34,50,66,67,68,69,70,71,72]. Chemically actuated MNMs function through multiple mechanisms, such as self-electrophoresis, self-diffusiophoresis (ionic self-diffusiophoresis/non-ionic diffusiophoresis), surface tension propulsion [73], bubble recoil, and self-thermophoresis.

### 3.1. Self-Electrophoresis

In recent years, significant attention has been given to the propulsion mechanism of self-electrophoresis in various MNMs, such as nanowires, nanorods, and Janus particles. Self-electrophoresis refers to the movement of MNMs induced by the creation of a localized electric field, which arises from the uneven distribution of ionic products [74]. This propulsion process has been the subject of numerous studies, with detailed discussions of its principles found in earlier reviews [75]. Self-electrophoresis allows MNMs to achieve autonomous motion through the exploitation of potential gradients, often necessitating the application of an external electric field [76]. For example, when Pt/Au nanowires are exposed to a H_2_O_2_ solution, the decomposition of H_2_O_2_ generates a concentration gradient of H^+^ ions, leading to the formation of a localized electric field. Because of their negative charge, the nanowires move in the opposite direction of this field, migrating toward the Pt end of the wire [77]. Building on this principle, Sen et al. introduced an innovative, bubble-free, asymmetric Pt/Cu nanowire capable of autonomous movement in dilute Br_2_ or I_2_ solutions [78]. The propulsion speed of this nanowire was found to increase with both the current density and the concentration of Br_2_/I_2_, while it decreased with an increase in the nanowire length. The current density is influenced by the redox potentials of the two metals, highlighting the importance of choosing distinct metals for optimizing the design [79]. Interestingly, as Br_2_/I_2_ is consumed, a short-circuit current is generated, which leads to a stronger redox effect in the Pt/Cu nanowire compared to traditional Pt/Au nanowires driven by H_2_O_2_.

Electrophoretic motion describes the movement of MNMs through a fluid under the influence of an applied electric field. In the case of self-electrophoresis, the autonomous propulsion of MNMs relies on the combination of at least two different metals within the nanostructure, forming a self-sustaining electrochemical system. For example, in Pt/Au nanorods, Pt acts as the anode, while Au serves as the cathode, enabling the rods to propel themselves (Figure 6). On the Pt side, the decomposition of H_2_O_2_ results in the generation of electrons (negatively charged) and H^+^ ions (positively charged). The H^+^ ions migrate along the bilayer surrounding the nanorods, while the electrons travel internally from the Pt to the Au side. This flow of electrons facilitates the reduction of H_2_O_2_ into H_2_O and O_2_ on the Au side. As a result, the movement of electrons from Pt to Au, along with the corresponding shift of H^+^ ions, creates a H^+^ concentration gradient along the axis of the nanorod. This electrochemical process imparts a negative charge to the nanorods, causing them to move toward the H^+^-rich region that was previously near the Au side.

In self-electrophoresis, charged MNMs are propelled forward by a self-generated electric field, which arises from the inhomogeneous distribution of ions. The velocity (*V*) of the MNMs is related to the self-induced electric field (*E*), the zeta potential of the MNMs (ζ), the permittivity (*ε*), and the viscosity of the medium (*μ*), as shown below:$$V=\frac{\zeta \varepsilon E}{\mu }$$

### 3.2. Self-Diffusiophoresis

Self-diffusiophoresis is a widely observed propulsion mechanism in which the motion of MNMs is induced by concentration gradients resulting from chemical reactions (Figure 7a). This phenomenon emerges from the uneven distribution of catalytic by-products around the MNMs, leading to a concentration gradient that drives their movement [80]. Self-diffusiophoresis can be divided into two distinct categories: ionic self-diffusiophoresis (Figure 7b) and non-ionic diffusiophoresis (Figure 7c), based on the nature of the solute involved [81]. The primary difference between these two types lies in whether or not the solute contains an electrolyte. In the case of ionic self-diffusiophoresis, charged solute molecules migrate within an electric field generated by ion concentration gradients, with the molecules moving toward regions of lower concentration. This results in the autonomous propulsion of MNMs [82]. However, this motion tends to be slow and highly dependent on the ionic strength and pH of the surrounding solution. In ionic self-diffusiophoresis, two mechanisms, electrophoresis and chemical swimming, occur simultaneously [83]. The varying diffusion rates of anions and cations within the solution generate localized electric fields. Additionally, the charged substrate creates an electroosmotic flow, which, combined with the local electric field, results in electrophoresis [84]. Further investigation and modeling are required to clarify which of these mechanisms predominates. On the other hand, non-ionic diffusiophoresis, which is simpler in nature, always occurs toward regions of higher ion concentration [85]. This type of propulsion relies on the changes in concentration gradients of neutral solute molecules to drive the movement of MNMs, but the generated force and speed are significantly lower compared to ionic self-diffusiophoresis [86].

The self-diffusiophoresis propulsion mechanism is commonly applied to spherical MNMs [50,67]. In this process, the catalyst, often Pt, is placed asymmetrically on one side of the MNMs. As the chemical reaction takes place, the by-products, H_2_O and O_2_, accumulate more abundantly near the catalyst, leading to the formation of a concentration gradient along the surface of the MNMs. Once the concentration of these products reaches a critical level, the increased local concentration drives the diffusion of the reaction products away from the catalyst. This movement generates a force that propels the MNMs forward.

### 3.3. Interfacial Tension

The imbalance of forces created by surface tension gradients along interfaces can induce fluid flow, a phenomenon referred to as the “Marangoni effect”. This mechanism, initially proposed by Crespi, Mallouk, and Sen, was used to model the propulsion of MNMs (Figure 8) [68]. In surface tension propulsion, the variation in surface tension magnitudes generates movement; however, the resulting force is typically low and highly sensitive to the specific surfactant used [87]. For instance, when H_2_O_2_ decomposes into H_2_O and O_2_ on the Pt side of Au/Pt nanorods, the increased production of O_2_ reduces the interfacial tension near the Pt surface. This difference in surface tension between the Pt and Au sides generates a force that propels the MNMs forward.

In the surface tension gradient model, the velocity of motion (v) of the MNMs is directly proportional to the surface tension (γ) of the solution, as described by the following equation:$$v=\frac{SR^{2}\gamma }{\mu DL}\propto k\gamma$$
where S, R, γ, μ, D, L, and k are the O_2_ generation rate, radius of MNMs, surface tension of solution, viscosity of solution, diffusion coefficient and length of MNMs, and constant, respectively.

### 3.4. Bubble Propulsion

The bubble propulsion mechanism, introduced as the initial model for understanding and designing the movement of MNMs, continues to be the most prevalent method in this area of research [88]. It relies on a redox reaction facilitated by a surface catalyst, resulting in the production of gas bubbles. As these bubbles detach from the MNM surface, they generate a momentum transfer that drives the MNMs forward [89]. Nevertheless, this method of propulsion is contingent upon gas release, which can affect the surrounding pressure and notably impact the movement characteristics of the MNMs [90].

Bubble propulsion has become a prominent mechanism in the exploration of MNMs. With the appropriate catalyst, MNMs of any shape can be activated using this method. The propulsion is initiated by the generation of microbubbles, which are produced through the catalytic breakdown of chemical fuels. Among the various MNMs employing bubble propulsion, those modified with Pt catalysts have been the most extensively investigated. These MNMs achieve autonomous motion through the decomposition of H_2_O_2_ into H_2_O and O_2_ bubbles (Figure 9) [71,72]. For instance, after the groundbreaking discovery of bubble propulsion by Whitesides et al., Pt was integrated into PDMS semicylindrical plates. The catalytic breakdown of H_2_O_2_ by Pt resulted in the formation of O_2_ bubbles, thereby enabling autonomous movement [91]. This seminal study set the stage for further research, leading to significant advancements in bubble-driven MNMs. Zhao et al. subsequently developed a set of detailed models for bubble propulsion, demonstrating that the velocity of the MNMs is influenced by both the surface tension of the liquid and the concentration of H_2_O_2_ [92]. The use of bubble propulsion markedly increases the power and efficiency of MNMs, highlighting their potential for transporting diverse payloads [93,94].

Bubble propulsion occurs when chemical fuels undergo spontaneous decomposition, forming micron-sized bubbles that are catalytically induced. These tiny bubbles detach from the MNMs’ surface, creating a recoil force that propels the MNMs away from the catalyst. Despite their small size, typically ranging from a few microns to several hundred microns, MNMs driven by bubble propulsion are capable of achieving high speeds and substantial propulsion forces. The speed of such MNMs has been a critical area of investigation due to its impact on motion efficiency. Solovev et al. demonstrated that the velocity of bubble-driven MNMs is influenced by both the frequency and size of bubble detachment. Their experiments revealed that collisions between bubbles caused significant measurement deviations, limiting the frequency of bubble separation and restricting their travel distance within the medium. As a result, the movement dynamics of MNMs were found to be affected by factors such as their shape, the composition of the fuel, and the viscosity of the surrounding fluid [71].

### 3.5. Self-Thermophoresis

Temperature gradients can drive the movement of MNMs through a mechanism called thermophoresis, also referred to as the “Soret effect” (Figure 10). This phenomenon is highly sensitive to temperature variations, making precise thermal control essential for maintaining stability [95]. The self-thermophoretic motion of MNMs was first investigated by Jiang et al., who studied its effects on individual particles [69]. Their experiments involved irradiating SiO_2_ Janus microspheres, with one hemisphere coated in Au, using an unfocused laser beam at a wavelength of 1064 nm. The Au layer absorbed the laser energy, generating localized heat, which in turn created a temperature gradient of approximately 2K across the particles. This thermal gradient was responsible for driving the self-thermophoretic motion of the microspheres.

### 3.6. Summary

MNMs are propelled through various mechanisms, which can generally be categorized into two main types: chemically driven and externally driven. Each category offers distinct advantages depending on the specific application (Table 2). Chemically driven MNMs harness chemical energy from fuels to propel themselves autonomously. Common fuels for this propulsion include H_2_O_2_, Br_2_, I_2_ solutions, hydrazine, and both acidic and alkaline solutions. Additionally, H_2_O is emerging as a clean and sustainable energy source for driving these systems. One of the key benefits of chemically propelled MNMs is the simplicity of their operation, alongside the wide variety of chemical reactions that can be employed for propulsion. However, these systems are significantly influenced by Brownian motion, which limits their controllability. Moreover, the dependence on potentially hazardous chemical fuels restricts their operational lifespan. As a result, choosing an appropriate propulsion method involves careful consideration of the MNMs’ operational environment, desired precision of control, and the nature of the tasks to be completed. This section reviews the five primary propulsion mechanisms identified in the literature for enabling autonomous movement in catalytic MNMs. It is evident that, when comparing similar systems, the propulsion mechanism is influenced not only by the shape of the MNMs but also by the concentration and characteristics of the fuel. In many instances, multiple mechanisms work simultaneously to drive propulsion. Therefore, further investigation is necessary to identify the individual contributions of each mechanism. A deeper understanding of these contributions will enable the optimization of propulsion systems, such as those based on bubble propulsion and electrophoresis, to achieve more precise control in practical applications.

## 4. Influence Factors

Chemically propelled MNMs generate autonomous motion by converting chemical energy from their environment into mechanical work. However, this energy conversion process is typically inefficient, leading to suboptimal speeds. For instance, in the self-electrophoresis propulsion mechanism, energy loss occurs in four distinct stages (Figure 11a). The initial stage involves the chemical and non-chemical breakdown of fuels like H_2_O_2_ by catalysts such as Pt [78]. The second stage occurs when highly exergonic reactions create a potential difference across the MNM surface, leading to some energy loss [85]. The third stage of energy dissipation arises from the limited potential difference between the cathode and anode, which reduces the amount of electrochemical energy that can be converted into mechanical motion [77]. The fourth stage involves additional losses due to electroosmotic flow, which acts against the self-electrophoretic movement of the MNMs [96,97]. Similarly, in the bubble recoil mechanism used for MNM propulsion, energy loss occurs in two stages (Figure 11b). Initially, O_2_ must form sufficiently large bubbles to detach from the MNM surface, but only a small fraction (around 1%) of the energy is used for bubble expansion, with the rest being lost as heat. Secondly, at low Reynolds numbers, propulsion is limited to the moment of bubble release. The acceleration and deceleration times are extremely brief, lasting only microseconds, which results in a short-lived recoil motion [98]. In addition to these inherent inefficiencies, various external factors such as the shape of the MNMs, the viscosity of the surrounding medium, and the composition of the fuel also play significant roles in influencing their overall dynamics, as will be discussed in the following sections.

### 4.1. Geometric Constraint

The motion behavior of MNMs is significantly influenced by their geometry, particularly in terms of symmetry and size. For instance, consider microtubules: if the tubular MNMs have a symmetric cylindrical shape, the released bubbles will detach in a linear fashion, resulting in a straightforward motion of the microtubules. On the other hand, if the edges of the microtubule are irregular, the bubbles will be emitted at an angle, causing the motion to follow a curved, spiral, or circular path [52,99]. This example highlights how the geometry of MNMs directly impacts their motion trajectories. In geometries with axisymmetry, the force generated by bubble release or fluid flow generally follows the axisymmetric direction, leading to linear or simplified rotational motion. Conversely, asymmetric structures result in uneven thrust forces across different directions, compromising both the stability and precision of motion. Furthermore, the size of MNMs plays a crucial role in determining their kinematic behavior. As the size of the MNMs decreases, surface effects—such as surface tension and viscous drag—become more dominant, influencing the propulsion mechanism. Smaller MNMs are more vulnerable to hydrodynamic and viscous forces, which increases their sensitivity to irregularities or external disturbances at the microscale, thereby affecting their motion trajectories. Optimizing the geometry of MNMs is thus key to improving their kinematic stability and accuracy. Additionally, the surface texture of MNMs has a substantial impact on their motion performance. Features such as fine protrusions, grooves, or non-uniform charge distributions can modify the flow dynamics of the surrounding fluid, affecting the bubble release process. By enhancing the local structure, MNMs can achieve more complex motion patterns. Finally, the flexibility of MNMs is another vital factor that influences their movement. Flexible designs allow MNMs to conform to various shapes and navigate through different environments, significantly improving their maneuverability in complex settings.

### 4.2. Media Viscosity

In fluids characterized by low Reynolds numbers, the motion of MNMs is primarily governed by viscous forces, with the influence of inertia being negligible due to the minuscule size of the particles. Sanchez et al. observed that microtubules exhibited rapid motion under physiological conditions, where the viscosity of the surrounding medium was reduced by 50%. This change led to a transition in the microtubules’ motion from linear to circular trajectories [100]. Such modifications in the dynamics of MNMs have contributed to an improvement in their performance [101]. Lowering the viscosity of the medium enables microtubules to move more efficiently in reconstituted blood at physiological temperatures, whereas at 25 °C, their movement is significantly hindered [102]. Pumera and colleagues also explored microtubule motion across fluids with different Reynolds numbers [103]. Their research supported Sanchez et al.’s findings, showing that circular or non-linear motion is more prevalent in fluids with higher Reynolds numbers [100]. Moreover, the composition of the surrounding solution can influence the motility of microtubules. Various factors, such as blood metabolites [104], proteins [105], extracellular thiols [106], electrolytes [107], and the water source, can all impact the motility and overall performance of microtubules [108,109].

Catalytic MNMs typically rely on localized chemical reactions to produce propulsive forces, which generate concentration gradients in the surrounding fluid. These gradients create a phenomenon known as “chemical kinetic flow”, which drives the autonomous motion of the MNMs. However, in low Reynolds number fluids, the motion of catalytic MNMs tends to slow down, and the efficiency of propulsion as well as the motion pattern is strongly influenced by the fluid’s viscosity. As a result, MNMs may exhibit erratic, random wandering or follow curved paths [98,100]. To improve the kinematic performance of MNMs, Bai et al. employed a rotating magnetic field, which enhanced the motion of helical carbon MNMs with lengths less than 8 μm in such low Reynolds number environments [110]. Under the influence of a rotating magnetic field, MNMs can adopt two distinct modes of motion: translation and rolling. The velocity of translation is dependent on both the frequency and intensity of the rotating field, with a maximum speed of up to 40 μm/s being achieved. In fluids with low Reynolds numbers, MNMs often exhibit more complex motion patterns, especially when considering their mean-square displacement (MSD) characteristics [111,112]. Due to the dominance of viscous forces, the movement of these MNMs is often random and diffusive, in contrast to the rapid, directional motion seen in high Reynolds number environments. Initially, MNMs in low Reynolds number fluids may exhibit some directional movement for short periods, with their average velocity correlating positively with fuel concentration. However, over time, their motion tends to shift back to random wandering, accompanied by a significant increase in diffusion coefficients [111]. Designing catalytic MNMs for such environments requires attention to two key factors: maximizing the local catalytic effect and minimizing drag during motion through structural optimization. The use of optimized shapes, enhanced surface properties (such as superhydrophobic or superhydrophilic coatings), or more efficient catalysts can considerably enhance the propulsion efficiency of these motors [101,102].

### 4.3. Fuel Composition

The motion of microtubules is significantly affected by the two primary components of the fuel, H_2_O_2_ and surfactants. Typically, as the concentration of H_2_O_2_ increases, the velocity of microtubules also rises, reaching a plateau at higher concentrations. This behavior mirrors the kinetics described by models such as the Langmuir adsorption isotherm or the Michaelis–Menten equation, which are used to explain enzyme reaction rates. Surfactants serve two essential roles in the fuel solution: they promote interaction between the fuel and the catalytic inner layer, and they support the stable formation and sustained release of microbubbles. Various surfactants, including common soap, benzalkonium chloride (BCl), sodium cholate, Triton-X, and sodium dodecyl sulfate (SDS), have been used for these purposes [113]. A comprehensive study conducted by the research groups of Pumera and Sanchez examined how different types of surfactants—anionic, cationic, and non-ionic—affect microtubule motion [113,114]. For anionic surfactants, increasing their concentration (below the critical micelle concentration) improved the velocity of microtubules. On the other hand, cationic surfactants, when present in high concentrations, were found to reduce microtubule speed and, in some cases, could entirely halt their movement. Non-ionic surfactants led to a more stable effect, with a slower but consistent increase in velocity. The interaction between surfactants and the catalytic inner layer of the microtubules had varied effects on their motility [113,114]. Interestingly, Simmchen and Zhao et al. observed that self-propulsion of microtubules could still occur in 10 wt% and 5 wt% H_2_O_2_ solutions, respectively, even without the presence of surfactants. However, these concentrations of H_2_O_2_ were approximately an order of magnitude higher than those typically used in surfactant-added fuel solutions [54,114].

### 4.4. Summary

The motion of MNMs is influenced by a range of factors, with three of the most significant being geometric constraints, fluid viscosity, and fuel composition. These variables not only affect MNM performance in isolation but also interact in ways that shape the motor’s trajectory, velocity, stability, and precision. Geometric constraints, such as shape, size, and surface structure, play a fundamental role in determining the propulsion mechanisms and movement paths of MNMs. Variations in these parameters lead to different kinematic outcomes. While asymmetric MNMs offer enhanced flexibility for specific practical uses, they often suffer from motion instability due to uneven propulsive forces, making control and navigation more challenging. Hence, it is crucial to balance flexibility with stability during the design process. Symmetrical designs are ideal for applications where stability is paramount, whereas asymmetric shapes are better suited for tasks requiring complex motion dynamics. The viscosity of the surrounding medium also significantly influences MNM motion by affecting the resistance the MNMs encounter. In more viscous environments, increased resistance can impede movement, whereas in low-viscosity media, MNMs are more susceptible to external fluid disturbances, which can destabilize their motion. In these cases, employing flexible or deformable structures may improve propulsion in high-viscosity environments. In contrast, increasing the interaction surface area between the MNMs and the fluid can enhance stability in low-viscosity settings. Finally, the fuel composition plays a crucial role in the propulsion of MNMs, as the energy output from chemical reactions is limited, and controlling these reactions presents challenges. Future advancements in MNM technology should aim at developing more efficient and controllable fuel systems, incorporating advanced catalysts and innovative energy storage or conversion methods, to extend the operational duration and enhance the overall performance of these MNMs.

## 5. Controlling Methods

The motion of MNMs in fluid environments is primarily governed by erratic Brownian motion, which poses significant challenges for their precise manipulation. Achieving controlled propulsion is therefore a critical area of research, as it directly influences the practical applicability of MNMs. In response to the limitations of chemical propulsion systems, such as poor controllability and limited operational lifetimes, the use of externally applied fields has emerged as a promising solution. Techniques involving external magnetic fields, electric fields, ultrasonic waves, and light have been explored to influence the motion of MNMs. These fields interact with the magnetic components, conductive surfaces, long axes, and sides of the MNMs, respectively, inducing effects like asymmetric charge distribution, torque generation, force imbalances, and photocatalytic reactions. Such interactions offer effective means to regulate both the speed and direction of MNM movement [115]. Recent research has showcased various strategies for achieving precise control over MNMs through the application of external fields (Table 3). This section provides an overview of the diverse methods employed to manipulate the propulsion of MNMs using these external influences.

### 5.1. Magnetic Control

One of the most common techniques for controlling and steering MNMs involves the use of an external magnetic field. By introducing paramagnetic or ferromagnetic materials into the MNMs during fabrication, these MNMs can be magnetized when exposed to such fields, allowing for controlled movement along a predetermined path. The incorporation of magnetic materials is typically achieved through processes like electrodeposition or PVD, with the choice of method depending on the desired geometry of the MNMs. Among the various materials used, Ni and Fe are the most frequently chosen due to their favorable magnetic properties.

Wang and colleagues presented a design for segmented MNMs featuring Au/Ni/Au structures that are driven by ultrasound and steered using an external magnetic field. To introduce structural asymmetry, a concave surface was created at the end of the Au segments through a spherical lithography technique. This alteration disrupted the symmetry of the design, contributing to the asymmetric nature of the MNMs. The interaction between the magnetic field and the intermediate Ni segments enables the MNMs to move with high precision and control. Furthermore, the Ni segments can also function as carriers for the transportation and delivery of magnetic particles (Figure 12A) [116].

Magnetic orientation has been proven to be a highly efficient approach for controlling the directional motion of self-assembled MNMs. A commonly employed method involves the deposition of a magnetic material onto a portion of spherical Janus MNMs through sputtering, which enables the MNMs to be influenced by an external magnetic field. For example, a 5 nm layer of Ni was applied to catalase-functionalized Janus capsule MNMs, followed by the addition of a Au layer. These biocatalytic Janus capsule MNMs demonstrated the ability to propel themselves in a H_2_O_2_-based cellular medium, with the magnetic field used solely for steering the MNMs towards target HeLa cells [117]. Importantly, the role of the magnetic field was limited to guiding the movement, not generating propulsion through magnetic attraction.

A commonly employed approach for fabricating self-propelled MNMs involves incorporating a Ni fraction into the structure of nanowires, nanorods, and microtubes through electrodeposition. A notable example is the Pt/Ni/Au/Ni/Au segmented nanowire, which was one of the first self-propelled designs [118]. The Ni component, due to its smaller size relative to the entire nanowire, results in transverse magnetization rather than longitudinal magnetization. This transverse alignment enables the nanowire’s magnetic moment to align with an external magnetic field, facilitating precise control over its movement within a fluid by adjusting the field direction. Experimental observations indicate that although the magnetic field can effectively guide the trajectory of the nanowires, it does not influence their speed. In a similar context, Burdick and colleagues demonstrated the directional propulsion of self-propelled Au/Ni/Au/Pt-CNT nanorods, which were also capable of transporting magnetic microbeads within a microfluidic system [37].

Magnetic steering of electrodeposited microtubes can be achieved by incorporating Ni through electrodeposition. In the case of conical microtubes, it is important to ensure that the Ni layer completely covers the inner surface before Pt deposition, enabling magnetization to occur along the tube’s axis. Alternatively, a simpler approach involves codepositing Ni and Pt to form a Ni/Pt alloy inner layer. This method imparts both magnetic and catalytic properties to the microtubes. However, using this alloy layer reduces the available catalytic surface area, which significantly diminishes the microtubes’ velocity when placed in dilute H_2_O_2_ solutions. In the case of striped microtubes with longitudinally arranged segments, the larger Ni portion along the axis allows for magnetization in the longitudinal direction, thus inducing motility characteristics similar to those of magnetotactic bacteria [119].

The incorporation of additional Fe layers during the deposition of rolled-up microtubes enables effective magnetic manipulation. When these microtubes are magnetized longitudinally, they are capable of aligning with and responding to an applied external magnetic field. Notably, Fe-containing magnetized microtubes have been shown to autonomously pick up and transport paramagnetic beads even without the need for an external magnetic field. Expanding on this concept, Sanchez and colleagues further demonstrated that Pt Janus particles can be used to transport cargo. To enhance both propulsion and the control of cargo delivery in catalytic Janus MNMs, they integrated multilayer Co/Pt magnetic caps onto the MNMs using PVD. These caps align the magnetic moments along the main axis of symmetry, allowing for precise manipulation of both the Janus MNMs and the superparamagnetic cargos under an external magnetic field (Figure 12B) [120]. Moreover, the magnetic manipulation of Janus particles was further showcased by their ability to sort microbeads into separate channels within a microchip device.

The ability to control microtube movement under an applied magnetic field has enabled their use as carriers for targeted transport. Despite the success of current methods, challenges persist in achieving precise manipulation and control of individual microtubes. Future studies are likely to explore closed-loop control systems and techniques for three-dimensional motion guidance. Notably, teams led by Misra and Sanchez have developed a highly accurate closed-loop control system for microtubes [121]. Using a weak magnetic field of 2 mT, they were able to control microtubes with high precision in a point-to-point manner. Another significant advancement was demonstrated in a separate study, where microtube movement was accurately controlled in both directions along and against the airflow. This was accomplished through an electromagnetic setup consisting of two orthogonal arrays of coils with Fe cores, coupled with two microscopy systems to direct the microtubes in three-dimensional space [121]. In this study, the microtubes successfully countered vertical forces, such as buoyancy and interactions with O_2_ bubbles, allowing them to swim upward or downward relative to a reference position. Magnetically responsive MNMs have proven capable of controlled movement within various environments when exposed to magnetic gradients and field moments, making them remotely controllable and retrievable [122]. However, their practical deployment remains hindered by intricate fabrication processes and bulky designs, which arise from their distinctive shapes and operational mechanisms [123].

### 5.2. Acoustic Control

Ultrasound (US) has demonstrated its ability to both power the motion of MNMs and act as an effective method for controlling their movement [124]. Over the years, numerous studies have explored the use of ultrasound for guiding MNMs and enabling rapid “stop/go” switching behavior in response to the activation or deactivation of the ultrasound signal. For ultrasound to effectively regulate the motion of MNMs, it is essential that the MNMs either have specific geometrical configurations or integrate a vibrating component that facilitates the additional movement [125]. Despite its potential, ultrasonic control is still limited by the need for specialized equipment to generate the ultrasound and the requirement for particular conditions to establish standing waves.

The motion direction of MNMs can be altered by adjusting the intensity of the ultrasound field. The swift and reversible shift between the aggregated and free-moving states of MNMs in H_2_O_2_ fuel occurs as a result of turning the ultrasound signal on and off (Figure 12C) [126]. The ultrasound field can also interfere with bubble generation. Wang et al. demonstrated reversible control over the propulsion of PEDOT/Ni/Pt microtubules by modifying the applied voltage to an external transducer that generates the ultrasonic field. Their study showed that the velocity of MNMs could change in less than 0.1 s, with “on/off” activation cycles that were faster and more reproducible than those achieved using other methods for halting MNM propulsion (Figure 12D) [127].

The integration of ultrasonic and chemically driven mechanisms to control reversible swarming and ultrafast motion of MNMs offers significant potential for a variety of applications, such as sensors, nanodevices, and drug delivery systems [19,128]. For instance, Wang et al. demonstrated that Pt-Au nanowire-based bubble-driven MNMs can rapidly adjust their speed in response to ultrasound, utilizing the pressure gradient generated by the ultrasonic field [128]. When the ultrasonic field is applied to the bubble-driven MNMs, the O_2_ produced within the catalytic tube is quickly expelled, eliminating the need for a growth process. The generated gases are then transported to the node or wave crest, inducing a swarming effect in the MNMs. Once the ultrasound is turned off, the MNMs disperse rapidly due to the autocatalytic reaction, causing the clustering behavior to cease. In a similar vein, ultrasound was used to control the motion speed of bubble-driven tubular PEDOT/MnO_2_ MNMs [19]. By using strong currents around resonant bubbles, ultrafast motion was achieved. The combination of ultrasound-induced and chemical actuation to regulate either ultrafast motion or clustering in MNMs is expected to be pivotal in advancing the development of biosensing technologies, nanodevices, and drug delivery systems that feature retrieval capabilities.

### 5.3. Electric Control

The method of controlling MNMs using electric fields offers the advantage of precise and rapid speed adjustment [129]. By combining direct current (DC) and alternating current (AC) electric fields, it becomes possible to direct, activate, and deactivate MNMs along a predetermined path while enabling fine-tuned control over their velocity. The primary role of DC fields is to regulate the movement speed of MNMs through electrophoretic and electro-osmotic forces. On the other hand, AC fields induce electrical torque that aligns the dipoles, thus governing the directional motion and orientation of the MNMs. For real-world applications, MNMs are required to navigate autonomously through complex fluidic environments. These environments are often characterized by non-uniform electric field distributions and varying conductivity, both of which can influence the motion and performance of the MNMs.

Metallic MNMs can be made to rotate with precision by applying AC voltages to multiple electrodes, which generate rotational torque (Figure 12E) [130]. Moreover, the movement of these MNMs can be controlled through dielectrophoretic forces when AC electric fields are applied to microelectrodes with specific configurations. These forces can drive metallic MNMs to form chains, accelerate their movement, and orient them along particular paths. Additionally, the MNMs can be manipulated to disperse, concentrate, or self-assemble into more intricate structures, enabling the creation of complex scaffolds (Figure 12F).

### 5.4. Light Control

Light, a readily available energy source, has been effectively employed to regulate the movement of MNMs [63]. Exposure to light induces asymmetric photocatalytic reactions on the MNMs’ surfaces, leading to the formation of charge gradients that drive autonomous motion through self-electrophoresis. Turning off the light source halts this motion [131]. However, this method has certain drawbacks, such as diminished light intensity at greater depths and reduced efficiency of self-electrophoresis in media with higher ionic strengths. For instance, Solovev et al. showcased the manipulation of microtubule motion using white light (Figure 12G) [132]. In their approach, the fuel solution was irradiated above a Pt-patterned Si surface, which lowered the local concentrations of H_2_O_2_ and surfactants. While white light can stop the microtubules’ movement, the researchers observed that shorter wavelengths more effectively inhibit microbubble formation compared to longer wavelengths. Additionally, the use of light to control the motion of microtubules is reversible: by dimming the light, previously inactive microtubules can be reactivated. However, the transition from active to inactive states is not immediate, with a delay of several seconds before the motion either ceases completely or stabilizes at its maximum velocity.

In a separate study, Wang and collaborators introduced oral cell-based MNMs that are powered by the conversion of H_2_O_2_ and controlled via near-infrared (NIR) light, allowing their motion to be initiated or halted. These MNMs demonstrate collective behaviors, such as aggregation or dispersion, which can be modulated by the presence or absence of NIR light, enabling precise control over their swarm dynamics [133]. Furthermore, these MNMs are capable of autonomously detecting H_2_O_2_ released by cancer cells and subsequently targeting and ablating these cells through a photothermal mechanism (Figure 12H).
Figure 12Motion Control for MNMs. (**A**,**B**) Controlling the motion of MNMs by using magnetic field. (**A**) Schematic diagram of an Au–Ni–Au metal alloy propelled by ultrasound and steered by the magnetic field. Reproduced from Ref. [116]. Copyright 2013, the American Chemical Society. (**B**) Scheme representing the magnetic steering of Janus micromotors. Reproduced from Ref. [120]. Copyright 2012, the American Chemical Society. (**C**,**D**) Controlling the motion of MNMs by using ultrasound. (**C**) Scheme representing controlling of acoustically propelled nanowire toward a HeLa cell. Reproduced from Ref. [126]. Copyright 2013, the American Chemical Society. (**D**) Scheme representing ultrasound-modulated bubble propulsion of chemically powered microtubes. Reproduced from Ref. [127]. Copyright 2014, the American Chemical Society. (**E**,**F**) Rotation of micro-/nanomotors by applying AC voltages to multiple electrodes: (**E**) schematic diagram of experimental setup of quadruple electrodes. (**F**) Images of one end fixed (**left**) and free (**right**) rotating Au nanowires. Reproduced from Ref. [130]. Copyright 2005, The American Physical Society. (**G**,**H**) Controlling the motion of MNMs by using light. (**G**) Switching the propulsion of individual m-engines off (**a**) and on (**b**) using a white-light source. Reproduced from Ref. [132]. Copyright 2011, Wiley-VCH. (**H**) Schematic cartoon for the surface plasmon resonance effect of Au/TiO_2_ under visible light. Reproduced from Ref. [133]. Copyright 2018, Wiley-VCH.
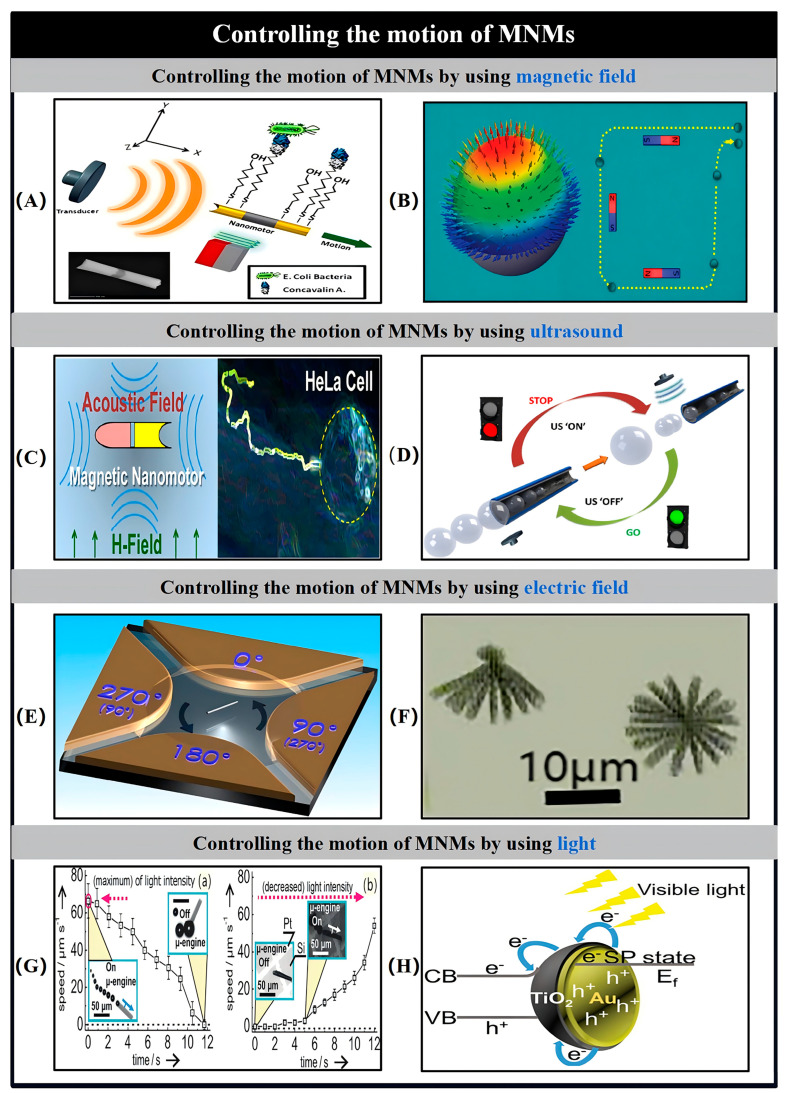


### 5.5. Thermal Control

Thermal control has proven to be an effective method for regulating the propulsion of artificial MNMs, including both nanowires and microtubules [134]. Elevated temperatures have been found to significantly increase the propulsion speed of Pt/Au nanowires. Likewise, a similar enhancement in motion is observed in bubble-driven microtubules, which are frequently employed to counteract the negative impact of limited fuel availability on propulsion efficiency.

The temperature of the solution can be controlled using two Peltier elements connected to a DC power supply, which are placed beneath the sample containing the microtubules. By heating the system to physiological temperatures, the propulsion efficiency of the microtubules is significantly improved, and movement can be achieved even in environments with low concentrations of H_2_O_2_ fuel (Figure 13A) [135]. Furthermore, flexible MNMs made from thermo-responsive polymer microjets can adapt to temperature variations in the surrounding solution. These microjets are capable of reversibly folding and unfolding with high precision, allowing for multiple activation and deactivation cycles as their shape changes, which in turn alters their radius of curvature. The use of stimuli-responsive materials in these systems offers a promising avenue for the advancement of intelligent MNMs.

### 5.6. Chemical Control

The propulsion of self-propelled MNMs can be regulated by adjusting fuel concentrations or introducing chemical stimuli. When exposed to a fuel concentration gradient, these MNMs exhibit the ability to move toward areas with higher fuel concentrations. This dynamic behavior not only enables the monitoring of fuel distribution but also provides a means to guide and fine-tune the propulsion process. For example, Mathesh et al. developed enzyme-driven 2D MNMs using a simple technique based on soft nanostructures, which demonstrated autonomous movement at very low fuel concentrations (0.003% H_2_O_2_). As the fuel concentration increased from 1 mM to 5 mM, the velocity of the 2D MNMs rose from 3.5 ± 0.04 μm/s to 6.53 ± 0.68 μm/s. These MNMs displayed positive chemotaxis, enabling them to swim against gravity due to the buoyancy force of the solute. Furthermore, the catalytic MNMs were capable of efficiently degrading methylene blue dye, achieving up to 85% removal efficiency [136]. To efficiently assemble MOF NPs into robust MOFs without the need for mechanical stirring, Huang et al. employed a rapid and large-scale self-assembly process using the Pickering emulsion technique, resulting in Fe_3_O_4_@NH_2_-UiO-66 colloidosomes (FeUiOsomes). Through redox reactions, these MNMs exhibited exceptional motility (450 ± 180 μm/s) in a 5 wt% H_2_O_2_ solution. The bubble-propelled MNMs not only demonstrated efficient removal of heavy metal ions (91% for Cr^6+^) but also achieved 94% removal of methyl orange [137]. In addition to chemical fuel concentrations, the motion of MNMs can also be influenced by other chemical substances. For instance, the addition of Ag^+^ significantly increases the velocity of Pt/Au nanowires, likely due to the under-potential deposition of Ag^+^ on the MNM surfaces, which alters both their surface properties and catalytic performance. Similarly, the propulsion speed of Au/Pt-CNT MNMs has been enhanced by the presence of N_2_H_4_. Surfactants, particularly in bubble-propelled MNMs, play a crucial role in generating and stabilizing microbubbles, which are essential for maintaining mobility.

Solovev et al. demonstrated that microbubbles, which are larger in size and formed by a small cluster of microjets, generate both chemophoretic and capillary forces. These forces work together to attract additional microjets into the group [138]. Building upon this, Baraban et al. conducted a more advanced study to investigate the chemotactic behavior of Janus MNMs and microtubules within microfluidic channels under carefully controlled conditions. Their results indicated that both MNM types exhibited directional motion that aligned with the fuel gradient, while capillary forces did not seem to have any significant effect on the movement (Figure 13B) [139].

### 5.7. Electrochemical Control

Wang and colleagues [140] explored the use of an electrochemical approach to control the motion of catalytic nanowires. They demonstrated that by gradually adjusting the potential of an Au electrode near the nanowire within its native solution, the propulsion rate of the nanowire could be precisely controlled, shifting from negative to positive potentials. Specifically, as the electrode potential was incrementally increased from −0.4 V to +0.4 V, +0.6 V, and +1.0 V, the nanowire velocity decreased from 20 µm/s to 16, 10, and 4 µm/s, respectively. This change in motion was reversible, with the nanowire’s speed increasing when the potential was returned to more negative values. The underlying mechanism for this control is linked to variations in the local O_2_ concentration, which is produced at the positive electrode and consumed at the negative one (Figure 13C).

### 5.8. Design Control

Although the previously mentioned techniques allow for precise control over the speed and path of nanorod motion, more sophisticated applications require the integration of complex movement patterns into the nanorods. This can be achieved by incorporating asymmetric geometries, which enable the nanorods to exhibit rotational motion. Zhao and colleagues employed a geometric shadowing technique to apply a thin catalytic layer asymmetrically on one side of nanowires. When these nanowires were introduced to a H_2_O_2_ solution, they demonstrated rotational motion around a fixed point near one end [49]. Due to the random orientations of the nanowires in the solution, both clockwise and counterclockwise rotations were observed. To further enhance this effect, the researchers designed L-shaped nanowires by combining geometric shadowing with substrate rotation, resulting in rotational movement between the long and short axes in the solution. Similarly, Mirkin’s group utilized on-wire lithography to asymmetrically coat nanowires, exposing a catalytic surface on one side, which induced rotational motion in an H_2_O_2_ solution [141]. Building on this, Mallouk, Sen, and their team employed vapor deposition to apply Cr, SiO_2_, Cr, Au, and Pt layers to one side of Au/Ru nanowires. These bimetallic nanowires achieved an average rotational speed of 180 rpm in a 15% H_2_O_2_ solution, with peak speeds reaching 400 rpm (Figure 13D) [142].

By altering the geometry of spherical particles, it becomes possible to achieve precise control over the movement of Janus particles through changes in their structural configuration. For instance, the deposition of a TiO_2_ layer on a Pt half-coated self-propelled particle can induce rotational motion in the MNMs [143]. On the other hand, coating the TiO_2_ arm with a Pt layer, rather than applying it to the spherical structure, results in curved trajectories for the particles, demonstrating an innovative design approach with potential applications in MNMs [144]. Furthermore, self-propelled spherical dimers can exhibit either quasilinear or quasicircular movement patterns, depending on the relative sizes of the two constituent monomers [46].
Figure 13Motion Control for MNMs. (**A**) MNMs’ propulsion controlled by temperature. (**a**) Motion of microjets in PBS solution. (**b**) Snapshots of microjets and bubble tails at 25 and 37 °C, respectively. Reproduced from Ref. [135]. Copyright 2013, The Royal Society of Chemistry. (**B**) MNMs’ motion controlled by chemical gradient in microfluidic channel. (**a**) Spherical Janus MNMs deviating towards peroxide solution with catalytic sites (dark areas on the particles) facing different positions, (**b**) microtubes deviate slightly towards peroxide with small angles. Red arrows indicate the direction of the MNMs. Reproduced from Ref. [139]. Copyright 2013, Wiley-VCH. (**C**) MNMs’ motion controlled by electrochemistry. Reproduced from Ref. [140]. Copyright 2009, The Royal Society of Chemistry. (**D**) MNMs’ motion controlled by design. Reproduced from Ref. [142]. Copyright 2009, American Chemical Society. (**E**) MNMs’ motion controlled by boundary condition. Reproduced from Ref. [145]. Copyright 2014, The Royal Society of Chemistry.
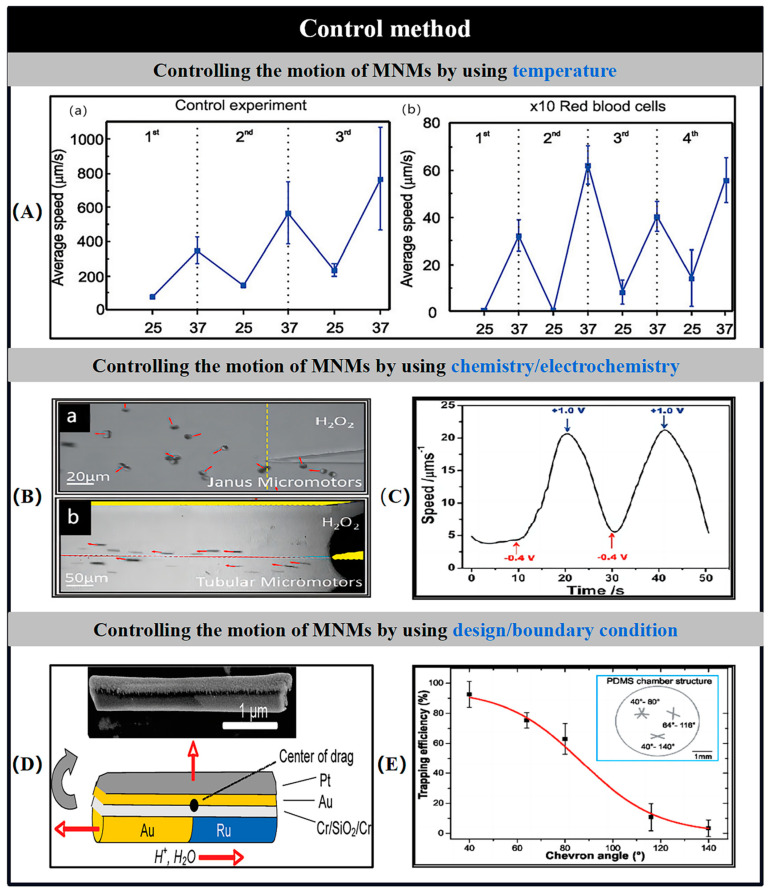


### 5.9. Boundary Control

Catalytic MNMs interact with surrounding surfaces, offering opportunities for novel control methods that do not depend on external energy inputs. Restrepo-Pérez et al. demonstrated a technique for capturing MNMs using steric boundaries designed into microfluidic chips, featuring chevron and heart-shaped structures (Figure 13E) [145]. The chevron structures, fabricated from PDMS with angles ranging from 40° to 140°, showed that larger angles led to better capture efficiency, in line with earlier theoretical predictions [146]. In addition, heart-shaped patterns and ratchet mechanisms have been used to sort and direct the motion of biological entities, such as E. coli and sperm cells [147,148,149,150]. Specifically, heart-shaped reservoirs with chevron tips at 40° angles are effective in concentrating microinjectors while preventing their backflow into the main reservoir. Ratchet mechanisms further enhance the retention of MNMs within chambers. This concentration method on microchips provides a versatile platform for capturing biofunctionalized microjets, facilitating easy integration into techniques for biomolecule concentration and the development of portable bioanalytical devices [151].

### 5.10. Summary

Catalytic MNMs can be controlled through a variety of methods, including acoustic, optical, magnetic, electrical, thermal, chemical, electrochemical, design-based, and boundary condition approaches. Each control strategy has its own set of advantages and limitations, which are contingent on the specific application. Acoustic control, for instance, enables non-contact manipulation via acoustic waves, allowing for precise positioning in complex liquid environments. However, its energy conversion efficiency tends to be low, and it is susceptible to interference from ambient noise. Optical control provides high spatial resolution, making it ideal for use in shallow, transparent media. On the other hand, its effectiveness diminishes in deeper environments due to rapid attenuation of light intensity, and the high energy consumption of light sources can lead to thermal damage. Magnetic control utilizes external magnetic fields to manipulate MNMs, making it particularly beneficial in biological contexts due to its non-toxicity and ability to be applied remotely. Nevertheless, its application is limited by magnetic field attenuation and reliance on magnetic materials, which can constrain its use in more complex settings. Electrical control offers precise directional control of MNMs through the manipulation of electric fields. However, it faces challenges related to the configuration of microelectrode arrays and potential electrolytic effects that may interfere with the operation. Thermal control, which relies on temperature gradients to regulate the movement of MNMs, is especially suitable for applications requiring sensitivity to ambient temperature changes. Yet, excessive heating can cause localized damage, making it less ideal for biological systems. Chemical and electrochemical methods harness reaction energy to regulate MNM velocity, offering a sustainable and self-sufficient approach. However, these methods can reduce the lifespan of MNMs due to fuel consumption and produce byproducts that may result in side effects. Design-based and boundary condition controls focus on optimizing MNM functionality by adjusting their shape, surface features, and the surrounding environmental conditions. While these methods provide valuable customization, their preparation is often complex and resource-intensive. A promising strategy for improving the precision and adaptability of MNM motion is the integration of multiple control methods or hybrid propulsion systems. Coupled with intelligent control systems, such integrated approaches can significantly enhance manipulation accuracy, broadening the potential applications of MNMs in fields such as biomedicine and environmental remediation.

## 6. MNM Motion Simulation Analysis

Although notable progress has been made in the development and control of various types of MNMs, the fundamental dynamics and physical principles that govern their movement remain inadequately studied. A deeper understanding of the physical characteristics of MNMs and the factors that influence their behavior is essential for optimizing their propulsion performance. Therefore, it is crucial to conduct extensive physical and hydrodynamic analyses of MNMs prior to their application in fields such as biomedical and environmental engineering. To investigate these dynamics, Wang et al. used nanowires as a representative case study [152]. A summary of the velocity expressions for MNMs driven by different mechanisms is presented in Table 4.

In the case of segmented catalytic nanowires, the structure is divided into two distinct parts, labeled as N and C, with their respective lengths represented by L_N_ and L_C_. Segment N is characterized as the non-catalytic portion, while segment C is the catalytic one. The model is placed in a solvent N (N = N_A_ + N_B_), with A and B denoting two distinct molecules in the catalytic environment. The volumes of A and B are defined as V = 32 × 32 × 32, and the molecular density is given as ρ = 9. The reaction occurring is A + C → B + C, which is irreversible. For this model, we assume that the probability of molecule A encountering the catalytic segment C is denoted by P_R_.

As shown in Figure 14A, within the indicated dashed region, it is assumed that there is a 99% likelihood of molecule A transforming into molecule B. Additionally, it is presumed that molecule B does not interact with solvent A, except during multiparticle collisions (MPCs). The time-dependent behavior of the system is described using a mesoscopic hybrid approach that combines molecular dynamics and multiparticle collision (MD-MPC) schemes [153,154,155,156].

As illustrated in Figure 14B, the nanowire is divided into discrete segments of length dldldl, with each segment being considered as a blob [155]. The distance between adjacent segments is assumed to remain fixed. Molecule A interacts with both segments N and C, experiencing Lennard-Jones (LJ) potential forces, with energy values ε_Cα_ (α = A, B) and separation distances σ_S_ (S = C, N).
(1)$$V_{C\alpha }\left(r\right)=4\varepsilon_{C\alpha }\left[{\left(\frac{\sigma_{S}}{r}\right)}^{12}-{\left(\frac{\sigma_{S}}{r}\right)}^{6}+\frac{1}{4}\right],\;r\leq r_{C}$$

For the analysis at hand, we assume a cut-off distance r_C_ = 2^1/6^σ_S_, where σ_S_ = 2. As shown in Equation (1), the molecule B in the solvent interacts with each component of the C segment through a repulsive LJ potential, maintaining identical energy parameters ε_CA_ = ε_NA_ = ε_CB_. In contrast, its interaction with elements of the N segment follows a different repulsive LJ potential, characterized by a distinct energy parameter (ε_NB_ = 0.1). The forces between the nanowires and the solvent molecules are then determined by differentiating the potential V.
(2)$$F=-V_{LJ}^{\prime }\left(r\right)=24\frac{\varepsilon_{S\alpha }}{\sigma_{S}}\left[{\left(\frac{\sigma_{S}}{r}\right)}^{7}-2(\frac{\sigma_{S}}{r}{)}^{13}\right],\;r\leq r_{C}$$

In the case of the considered segment S(i), the force F_S_(i) is the result of the combined interaction between the segment S(i) and the molecules A and B. Here, S can represent either the catalytic segment C or the non-catalytic segment N. As shown in Figure 14C(a,b), based on the principle of spatial force parallelism, the force F_S_(i) can be characterized by the force F_QS_(i), which acts through the center of mass, along with the torque M_QS_(i), which acts around the center of mass. The specific formulations for the force F_QS_(i) and the torque M_QS_(i) are provided below for further clarification.
(3)$$F_{QS}\left(i\right)=F_{S}\left(i\right)$$


(4)
$$M_{QS}\left(i\right)=\left[\left(\frac{\left(L_{N}-L_{C}\right)}{2}+(-1{)}^{k}idl\right)\right]e\times F_{S}\left(i\right)$$


Here, e denotes the unit vector that points from the non-catalytic section to the catalytic section. For the case where S = C, the value of k = 0; in all other cases, k takes the value of 1.

The total force and moment acting through the center of mass, as shown in Figure 14C(c), are defined by the following expressions:(5)$$\begin{array}{c} F_{Q}\left(i\right)=\sum_{i}^{n}F_{C}\left(i\right)+\sum_{i}^{m}\;F_{N}\left(i\right) \end{array}$$
(6)$$M_{Q}=\sum_{i}^{n}\;\left(\frac{\left(L_{N}-L_{C}\right)}{2}+idl\right)e\times F_{C}\left(i\right)+\begin{array}{c} \sum_{i}^{m}\;\left(\frac{\left(L_{N}-L_{C}\right)}{2}+idl\right)e\times F_{N}\left(i\right) \end{array}$$

When the nanowire is treated as a rigid object, the total force exerted on it can be separated into two components: one corresponding to the translation of the center of mass and the other to its rotation around the center of mass. Therefore, the position of the center of mass at a time increment t + dt can be computed using the Verlet algorithm, as outlined below:(7)$$r_{Q}\left(t+dt\right)=r_{Q}\left(t\right)+v_{Q}\left(t\right)dt+0.5dt^{2}\cdot \frac{F_{Q}\left(t\right)}{m_{CN}}$$

The mass of the nanowire segment is denoted as m_CN_ = ρ_L_·L_CN_, where ρ_L_ = 562.5. Subsequently, the velocity of the nanowire’s center of mass can be determined using the following equation:(8)$$v_{Q}\left(t+dt\right)=v_{Q}\left(t\right)+0.5dt\left(F_{Q}\left(t\right)\right)+\frac{F_{Q}\left(t+dt\right)}{m_{CN}}$$

Thus, Equation (6) can be expressed as:(9)$$M_{Q}=e\times g$$
(10)$$g=\sum_{i}^{n}\;(\frac{\left(L_{N}-L_{C}\right)}{2}+idl)F_{C}(i)+\begin{array}{c} \sum_{i}^{m}\;\left(\frac{\left(L_{N}-L_{C}\right)}{2}+idl\right)F_{N}(i) \end{array}$$

The combined moment around the nanowire’s center of mass can also be expressed as:(11)$$M_{Q}=e\times g^{⊥}$$
(12)$$g^{⊥}=g-\left(g\cdot e\right)e$$

Consequently, the rotation equation can be partitioned into two differential equations:(13)$$\dot{e}=u$$
(14)$$\dot{u}=\frac{g^{⊥}}{I}+\lambda e$$
(15)$$I=\frac{1}{4}m_{CN}R^{2}+\frac{1}{12}m_{CN}L_{CN}^{2}$$

The Verlet leap frog method allows the position of the nanowire at the moment t + dt to be expressed as follows:(16)$$r_{Si}\left(t+dt\right)=r_{Q}\left(t+dt\right)+\left(\frac{\left(L_{N}-L_{C}\right)}{2}+(-1{)}^{k}idl\right)e\left(t+dt\right)$$

When S = C, k = 0; otherwise k = 1.

The molecular dynamics of the solvent are modeled using standard molecular dynamics principles, excluding the nanowires. Throughout the simulation, the system is partitioned into a grid of cells. The time interval, denoted as τ, is employed to discretize the collision time for particles within a given cell. As a result, the position and velocity of particle I in cell I at a future time step, t + τ, can be computed using the Verlet algorithm, leading to the following expressions:(17)$$r_{i}\left(t+\tau \right)=r_{i}\left(t\right)+v_{i}\left(t\right)\tau +0.5\tau^{2}\cdot \frac{F_{i}\left(t\right)}{m_{i}}$$
(18)$$v_{i}\left(t+\tau \right)=v_{i}\left(t\right)+\frac{0.5\tau \left(F_{i}\left(t\right)+F_{i}\left(t+\tau \right)\right)}{m_{i}}$$

The velocity of particle I in post-collision can be expressed as:(19)$$v_{i}^{\prime }\left(t+\tau \right)=v_{cm,I}\left(t+\tau \right)+\hat{\omega }\left(\alpha \right)\left(v_{i}\left(t+\tau \right)-v_{cm,I}\left(t+\tau \right)\right)$$
(20)$$v_{cm,I}\left(t+\tau \right)=\frac{\sum_{i=1}^{N_{I}}\;v_{i}\left(t+\tau \right)}{N_{I}}$$

In this context, V_cm,I_(t + τ) refers to the velocity of the center of mass for the particles within cell I, while $\hat{\omega }\left(\alpha \right)$ represents the rotation operator assigned to each individual cell within the system. Additionally, N_I_ denotes the number of particles present in cell I.

To summarize, Equations (2)–(6) describe the position of the nanowire, while its velocity is represented by Equations (7) and (8). It is crucial to highlight that all parameters in the dimensionless LJ unit are scaled using the energy ε, the mass of the center of mass m, and the distance σ.

Self-propelled MNMs, often viewed as a form of advanced bionic material, stand out due to their ability to move independently, dynamically assemble, and exhibit collective behaviors. Leveraging numerical simulations offers valuable insights into the mechanisms underlying their motion and allows for a deeper exploration of interactions among multiple bodies.

In their research, Wang et al. utilized a two-dimensional axisymmetric model to simulate bimetallic nanowires [157]. To capture the chemical processes at the nanowire tips, they incorporated H^+^ inflow and outflow at these locations. The boundary conditions for the nanowires were specified as electro-osmotic flow boundaries. The self-electrophoretic mechanism of MNMs, discussed in Section 3.1, plays a key role in this system. The chemical reactions occurring at the ends of the nanowires result in an uneven distribution of charged ions, which in turn generates a self-induced electric field that propels the motion of the MNMs.

The self-electrophoretic mechanism involves a dynamic interaction among fluid motion, ion transport, and the electric field surrounding the nanowire. To simulate the ionic, electric, and flow fields around the nanowires, the approach effectively integrates dilute matter transport, electrostatic interactions, and peristaltic flow dynamics. Figure 14D outlines the core computational principles used in the simulation, while Figure 14E illustrates the boundary conditions applied during the simulation process [152].

The simulation results revealed that the flow field around the bimetallic nanowire shows the Pt section moving in a forward direction, while the surrounding electric field exhibits dipole-like characteristics. To model these physical processes, the simulation of the electric and flow fields was carried out by effectively coupling the interfaces of dilute matter transport, electrostatics, and creeping flow within COMSOL software 6.1 (Figure 14F) [157]. Additionally, the study examined the nanowire’s velocity and the interaction forces between two nanowires (Figure 14G) [157]. In simulations of catalytic MNMs, the local concentration gradient generated by the catalytic reactions plays a key role in determining the nanowires’ motion. The model quantifies the effects of local fluid flow by simulating both the consumption of reactants and the production of products at the catalyst surface. This process includes not only the changes in reactant concentration but also the phenomena of bubble formation, dissolution, thrust, and the diffusion of reaction products during the reaction. Under low Reynolds number conditions, inertial forces become negligible, causing the flow to be dominated by viscous effects. As a result, the motion of nanowire MNMs within the fluid is characterized by a high degree of randomness and diffusion. The simulations suggest that nanowire MNMs typically exhibit slow, random wandering, lacking stable, directional propulsion. Apart from hydrodynamic factors, the geometry, surface smoothness, and the hydrophilic or hydrophobic properties of the nanowires significantly influence their motion performance. For example, elongated nanowires display greater propulsion efficiency compared to spherical or other shapes, as the increased contact area with the fluid enhances the local flow dynamics. Further analysis reveals the impact of various design parameters (e.g., size, surface modification, catalyst material) on the nanowire MNMs’ performance. Smaller nanowires tend to move more slowly but are more flexible and better able to adapt to microscopic environments. In contrast, larger nanowires may generate greater propulsion force but exhibit less stability, making them more susceptible to disturbances in the fluid.

## 7. Application

Recent progress in load-bearing capabilities, propulsion mechanisms, and durability has significantly enhanced the potential of artificial MNMs, positioning them as promising candidates for a variety of applications in biomedicine (Figure 15) and environmental cleanup. These diverse applications span multiple domains, each necessitating customized functionalization approaches to address specific requirements. This section delves into the potential uses of MNMs in areas such as load transport, targeted drug delivery, disease diagnosis, and environmental remediation.

### 7.1. Cargo Delivery

Biomotor proteins play a crucial role in the intracellular transport of vesicles and organelles along microtubule filaments. This natural biological function has served as the basis for developing early applications of synthetic MNMs, designed for tasks such as picking up, transporting, and delivering model cargo systems, including nano- and microparticles [158]. For example, Pt/Au nanorods were modified by attaching them to polystyrene (PS)–amidine and PS–streptavidin microspheres through electrostatic interactions and biotin–streptavidin binding, respectively (Figure 16A). The self-electrophoresis mechanism enables these MNMs to transport particles of various sizes. However, an increase in particle size leads to higher viscous drag, resulting in a decrease in the propulsion speed of the MNMs. To improve their orientation control and magnetic load handling, Ni-segmented nanorods are employed [159,160]. The release of the transported particles is initiated by reversing the MNMs’ motion direction using an external magnetic field, which overcomes the forces holding the particles, thus enabling their release. Moreover, the propulsion velocity of MNMs within microfluidic channels is typically lower than that in open environments.

Another method for load release involves the inclusion of Ag segments positioned between the Pt and Au sections in Pt/Au/Ag/Au nanorods, or alternatively, the use of a photocleavable linker to attach biotin-functionalized cargo to Pt-Au-PPy-PPyCOOH nanorods [161]. In these configurations, the dissolution of the Ag segments initiates the efficient release of the cargo. However, these nanorods are constrained by their relatively low power output and suboptimal performance in electrolytic and ionic environments, which significantly limits their effectiveness in cargo delivery.

Spherical Pt/PS Janus particles are capable of aggregating with unmodified PS particles and facilitating their transport through van der Waals interactions. On the other hand, magnetic Pt/SiO_2_ Janus particles excel at manipulating paramagnetic microbeads [70,120]. The motion of these Janus particles can be accurately regulated within intricate microfluidic systems by applying magnetic fields. This precise control enables various processes such as selective cargo capture, particle sorting, and the aggregation of colloidal particles (Figure 16B).

Microtubules have demonstrated considerable promise in both targeted and non-targeted load transport applications. Non-targeted transport typically involves the movement of particles and cells in a forward direction [162,163]. For example, as shown in Figure 16C,D, microtubules are employed to transport cellular and colloidal particles, respectively. On the other hand, targeted delivery necessitates the modification of microtubules with biochemical recognition sites that specifically bind to designated targets such as cancer cells, proteins, bacteria, or sugars [164,165,166,167,168,169]. Furthermore, ferromagnetic microtubules have the ability to selectively capture and isolate paramagnetic microbeads from a mixture of particles, all without the need for surface modifications [170].

### 7.2. Drug Delivery

Administering individual drugs may result in toxic effects on the body during their transport and delivery, which could lead to adverse reactions. Additionally, the complex microenvironment surrounding the target site may alter the drug’s behavior, raising concerns related to both biocompatibility and safety. Improving the precision of targeted drug delivery systems can decrease the necessary dosage of therapeutic agents, partially mitigating the issue of high drug costs. However, certain drugs naturally encounter biological barriers that restrict their ability to be transported effectively. This highlights the urgent need for the design of intelligent drug delivery systems capable of preventing drug degradation, overcoming biological barriers, and enhancing bioavailability [171,172,173,174]. The development of appropriate MNMs for controlled release is crucial for achieving these goals. Within the biomaterials field, research is actively progressing towards novel drug delivery systems, holding significant potential for advancing therapeutic strategies and improving disease prevention techniques [175].

Chemically driven MNMs are capable of facilitating both the loading and controlled release of drugs by responding to specific environmental stimuli or sensitivity variations. For example, doxorubicin (DOX), a widely used chemotherapeutic agent, is encapsulated within Mg-based MNMs guided by photoacoustic computed tomography (PACT) for targeted gastrointestinal therapy [176]. Advanced microtubular engines, driven by bubbles, can efficiently load DOX via π–π stacking interactions and release it as needed, making them particularly valuable for bioanalytical applications [177]. Nanographene-based microengines convert chemical energy into motion, loading DOX through physical adsorption, with controlled release achieved via modifiable electrochemical processes (Figure 16E) [178]. Additionally, magnetic MNMs have demonstrated the ability to penetrate tumors deeply and deliver drugs intracellularly by incorporating antibodies with catalytic properties. These systems have shown success in inhibiting tumor spheroid growth in human colorectal carcinoma (HCT116) cells. Moreover, MNMs have exhibited remarkable mobility in serum, highlighting their potential for the development of effective therapeutic strategies [179]. Pt-mesoporous SiO_2_ Janus particles, loaded with smart drugs, have proven effective in treating THP-1 leukemia cells. The use of glutathione as a trigger enables self-propelled drug release, significantly improving cell internalization and enhancing therapeutic efficacy [180].

Ongoing research into chemically driven MNMs is continuously improving their capacity for drug loading and release. The propulsion mechanisms that power self-driven MNMs are particularly important for their application in the biomedical field. For example, the motion of Cu/Pt-catalyzed microtubules is significantly affected by key components in serum, with protein adsorption on their surfaces playing a major role in their behavior [105]. Supramolecular MNMs, which are self-guided, can achieve precise motion in response to varying fuel concentrations, enabling both autonomous and directed movements [181]. Polymeric MNMs can control their propulsion powered by H_2_O_2_ by adjusting the response rate to blue light exposure. Moreover, the strategic grafting of different materials allows for the creation of diffusion channels, which enables the regulation of drug release in real time [182]. Janus MNMs, which exhibit directional motion, offer a biocompatible and non-toxic approach for controlled drug delivery, ensuring targeted treatment at specific sites (Figure 16F). Additionally, the in vitro release of drugs can be effectively controlled by adjusting the surrounding pH, which directly influences the drug’s functional properties [183].

Bionically inspired MNMs, which include those derived from natural systems such as cell membranes, bacteria, and viruses, have shown great promise in drug delivery and release applications. These MNMs, when fully loaded with therapeutic agents, can follow programmed pathways to efficiently transport drugs to their designated targets. Upon reaching the target site, the membrane surrounding the MNMs dissolves, triggering the automatic release of the drug and the separation of the delivery vehicle. Once their mission is complete, these MNMs can degrade autonomously, supporting a variety of biomedical functions [184]. Chemically driven MNMs have also been developed by incorporating mesoporous SiO_2_ nanoparticles into neutrophils. The bacterial films coating these SiO_2_ spheres possess intrinsic chemotactic properties, which allow for stealthy and targeted drug delivery to specific locations (Figure 16G) [185]. Moreover, chemically driven MNMs with plasmonic properties, designed in the shape of virus-sized rectangular particles, enable precise tracking and localization of subcellular components (Figure 16H). These MNMs are also capable of delivering protein payloads to prostate cancer cells, a type of cell that is generally resistant to transfection, offering a new avenue for protein-based therapies [186]. This exploration of drug loading and release using artificial, hemispherical, and bionically inspired MNMs emphasizes their promising potential for future applications in practical biomedical fields.
Figure 16Potential applications of MNMs. (**A**–**D**) Chemically driven MNMs for load transport. (**A**). Left: Cargo attachment to the nanorods by electrostatic interaction between the negative PPy end of Pt–Au–PPy MNMs and a positively charged PS–amidine microsphere; right: biotin–streptavidin binding between the Au tips of Pt–Au rods functionalized with a biotin-terminated disulfide and streptavidin-coated cargo. Reproduced from Ref. [158]. Copyright 2008, American Chemical Society. (**B**) Manipulation of colloidal cargo in microfluidic channels using magnetic Pt/SiO_2_ micromotors. Reproduced from Ref. [120]. Copyright 2012, American Chemical Society. (**C**) Manipulation of neuronal CAD cells (cathecolaminergic cell line) by rolled-up Ti/Fe/Pt microtubes. (A) MNMs directed towards the CAD cell, its transport (B) and delivery in a desired location by a quick rotation of the magnet (C). Reproduced from Ref. [163]. Copyright 2011, Royal Society of Chemistry. (**D**) Loading and transport of 5 µm PS cargo particles by the Ti/Fe/Pt MNMs in a microchannel. (A) Moving MNMs sorting a residual bubble into a 150 μm wide microchannel in a PDMS microchip. (B, C) Zoomed images of the MNMs loading (B) one and (C) two microparticles. (D, E) MNMs transporting (D) one and (E) two microparticles into the microchannels.Reproduced from Ref. [162]. Copyright 2011, American Chemical Society. (**E**–**H**) Chemically driven MNMs for drug delivery. (**E**) Loading of doxorubicin (DOX) on the n-rGO/Pt micromachines. (a) The molecular structure of doxorubicin. The conjugated side is highlighted. (b) Schematic illustration of n-rGO/Pt micromachines loaded with DOX. Reproduced from Ref. [178]. Copyright 2019, Wiley-VCH. (**F**) Schematic representation of the fabrication process for Mg–Au–Drug–Polymer MNMs. Reproduced from Ref. [183]. Copyright 2022, Elsevier BV. (**G**) Chemotactic motion of hybrid neutrophil MNMs toward a gel containing E. coli in a microfluidic channel. Reproduced from Ref. [185]. Copyright 2017, Wiley-VCH GmbH. (**H**) Internalization of Protein Cargo Carried by Au/Ag MNMs. Reproduced from Ref. [186]. Copyright 2021, American Chemical Society.
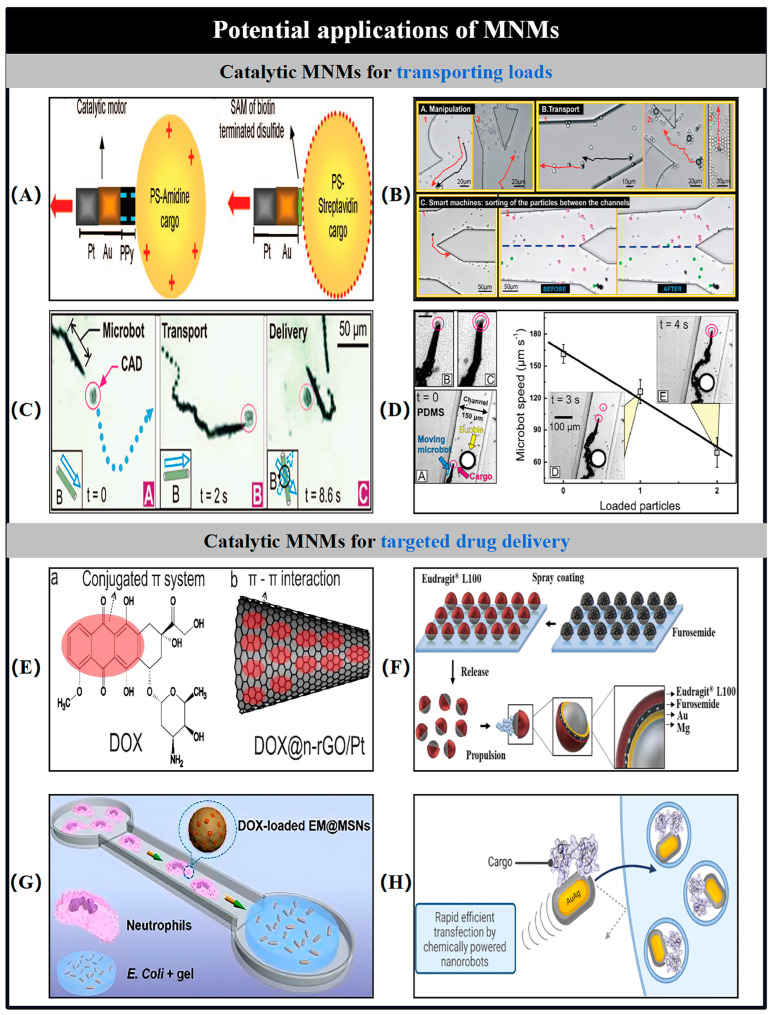


### 7.3. Disease Diagnosis

The process of “disease diagnosis” involves employing various technical methods to identify a disease by analyzing specific biomarkers or indicators. Initially, a preliminary diagnosis is made through the careful assessment, synthesis, and evaluation of the patient’s symptoms and medical history. This initial diagnosis is then validated and refined to ensure its precision and reliability. The accuracy of diagnostic results is typically measured through key metrics such as sensitivity, specificity, and overall diagnostic accuracy. Moreover, a comprehensive diagnosis is often complemented by medical impact assessments and additional clinical tests that aid in forming a complete understanding of the patient’s condition [187,188,189]. From the perspective of chemically driven MNMs, diagnostic MNMs can be strategically engineered to achieve precise diagnostic objectives, enabling targeted outcomes for specific medical conditions.

Chemically powered MNMs have emerged as powerful tools for diagnostics and detection within the life sciences. These catalytic MNMs are predominantly employed for the intracellular identification of small molecules, biomarkers, and other important macromolecules. For example, by utilizing established structural frameworks and functionalities, enzyme-driven MNM devices can operate at varying speeds, enabling enzymatic assembly processes for DNA hybridization. The high mobility exhibited at low H_2_O_2_ concentrations allows for the release of multiple molecules from target DNA, thereby improving the sensitivity of assays [190]. Additionally, self-sensing enzyme-driven MNMs can detect changes in environmental pH and monitor their own activity, functioning as indicators of their operational state. This ability enhances the evaluation of MNM performance across different environments and applications (Figure 17A) [191]. Furthermore, palindromic DNA hairpin MNMs can undergo rapid self-assembly and initiate transcription, greatly improving the accuracy of clinical sample detection [192].

Smaller biomarkers can be effectively detected using Au/Pt bimetallic tubular MNMs, which lower the detection threshold for biomarkers such as miRNA-21, thereby improving the performance of these miniature sensing systems [193]. Similarly, Janus mesoporous microsphere/Pt-based MNMs accelerate the detection and identification of microRNAs in complex biological samples. The recognition system on their surface promotes specific receptor–target interactions, providing an innovative method for analyzing miRNAs in these samples (Figure 17B) [194]. The fluorescence characteristics of these intelligent autonomous systems, which can be clearly distinguished under UV light excitation, make it possible to differentiate them from surrounding mixtures. This enables a novel form of self-driven detection, where MNMs can be employed for diverse identification tasks in complex environments [195]. Furthermore, Mg/Pt Janus MNMs have been used to enhance the sensitivity of glucose detection in human serum. These MNMs correlate the current signal with glucose concentration, improving the glucose detection limit (Figure 17C) [196]. Pt-based MNMs also serve as efficient probes for detecting bacterial toxins in fluorescence assays and for removing various substances in bioassay applications [197].

Chemically driven plasma virus-sized MNMs are capable of efficiently transfecting proteins within minutes, achieving up to six times the efficiency of traditional methods. The plasma’s inherent properties enable the precise tracking and localization of subcellular components, a key factor in the protein transfection process [186]. Additionally, the MNM-based bead-motion cellphone (NBC) system offers the ability to detect variations in Zika virus concentration and motion-related changes, making it a highly specific tool for pathogen detection and disease management (Figure 17D) [198]. Furthermore, plasma-powered self-propelled MNMs have been shown to enhance immune responses in living organisms [199]. In diagnostic applications, chemically driven MNMs demonstrate exceptional sensitivity and accuracy, offering valuable insights into the microenvironment by modifying detection indicators, both on their surface and internally. These advancements have significantly propelled the development of MNMs for diagnostic purposes.
Figure 17Potential applications of MNMs. (**A**–**D**) Chemically driven MNMs for disease diagnosis. (**A**) Self-sensing enzyme-powered MNMs equipped with pH-responsive DNA nanoswitches for disease diagnosis. Reproduced from Ref. [191]. Copyright 2019, American Chemical Society. (**B**) Schematic diagram of meso-MS/Pt/DNA MNMs for miRNA detection in vitro. Reproduced from Ref. [194]. Copyright 2022, American Chemical Society. (**C**) Schematic representation of Mg/Pt-Janus-MNM-assisted glucose biosensing in human serum using SPEa. Reproduced from Ref. [196]. Copyright 2019, American Chemical Society. (**D**) Schematic of the NBC system for virus detection. Reproduced from Ref. [198]. Copyright 2018, American Chemical Society.
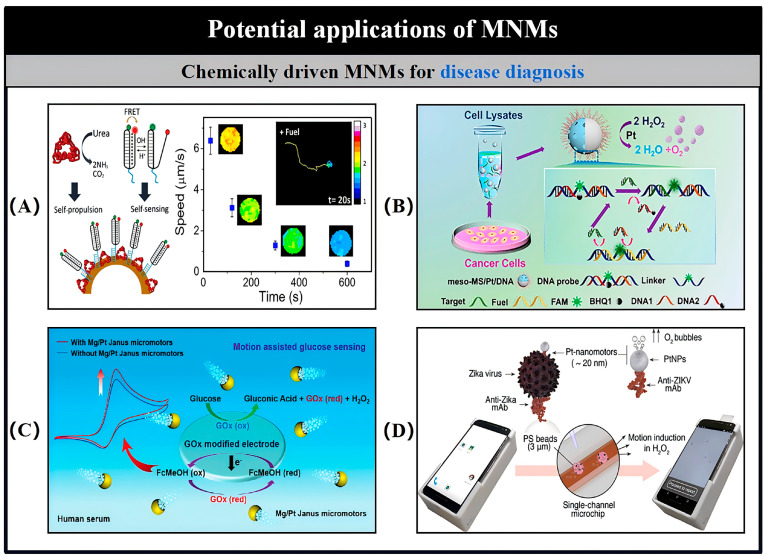


### 7.4. Environmental Remediation

The rapid expansion of global industrial activities has resulted in the unchecked release of hazardous substances into the environment, significantly threatening aquatic ecosystems. Disruptions in the availability of clean water could potentially impact millions of lives, raising substantial concerns across scientific, economic, and political spheres. Industrial processes generate a wide range of toxic pollutants, including organic solvents, pharmaceuticals, personal care and consumer products, pesticides, heavy metals, and various excipients and additives used in product formulations. Nanotechnology and nanomaterials present innovative approaches for the detection and removal of these contaminants, owing to their unique optical and catalytic properties, large surface area, and versatile surface chemistry [200]. The development of next-generation water purification technologies aims to eliminate both biological and chemical pollutants through methods that are cost-effective and energy-efficient, a need that is particularly critical in remote regions where conventional approaches may not be viable. Chemically driven MNMs are particularly suited for environmental remediation due to their exceptional catalytic abilities, self-propulsion mechanisms, and the capacity to enhance diffusion-limited processes (Figure 18). MNMs’ environmental applications can generally be categorized into two broad areas: the detection and removal of contaminants.

In environmental sensing applications, the altered movement patterns of MNMs in response to specific pollutants present an intriguing phenomenon that warrants further exploration. For instance, Pt/Au bimetallic nanowires demonstrate a noticeable increase in speed when exposed to Ag^+^ ions (Figure 19A) [201]. The interaction between Ag^+^ and the Pt segments results in the reduction of Ag^+^ to metallic Ag in the presence of H_2_O_2_, which enhances the electrocatalytic efficiency of the MNMs, thereby accelerating their movement. This enhanced locomotion is directly correlated with the concentration of Ag^+^, enabling the use of MNMs for real-time, quantitative toxicity assessments of Ag^+^ at nanomolar concentrations. On the other hand, pollutants can have a detrimental effect on MNMs’ motility, either by inhibiting or poisoning their catalytic activity [202]. For example, catalase-based MNMs show significant changes in their locomotor behavior when exposed to various inorganic and organic pollutants that interfere with enzyme function (Figure 19B) [203]. By evaluating common ecotoxicological metrics, such as exposure concentration (EC_50_—the concentration causing a 50% reduction in microfish locomotion) and life expectancy, it has been demonstrated that MNMs’ locomotion can be tracked optically in real time to assess water quality. Additionally, MNMs can respond rapidly to vapor plumes of chemical warfare agents (CWAs), enabling fast detection with minimal delay [204]. In the absence of CWAs, catalase-based MNMs maintain high activity levels; however, in their presence, enzyme activity is inhibited, resulting in a decrease in MNM mobility. Some contaminants are known to reduce the fluorescence intensity of polluted water. To address this issue, researchers have developed MNMs with fluorescent properties that allow for the detection of specific organic or inorganic contaminants [205,206]. Furthermore, MNM-assisted fluid mixing and accelerated chemical degradation of organophosphate (OP) threats have facilitated the creation of a rapid electrochemical monitoring method for decontamination processes [207]. For example, the conversion of paraoxon (a non-detectable substance) to p-nitrophenol was monitored using printed electrochemical sensor strips. The miniaturized and self-propelled nature of MNMs makes them highly effective for environmental monitoring and water treatment. However, producing dual-functional MNMs that can simultaneously detect and remove organic pollutants from water remains a significant challenge. Xing et al. tackled this issue by using calcined MgAl-layered double hydroxide (MgAl-CLDH) nanosheets combined with Co_3_O_4_-C nanoparticles to design a novel type of Janus MNM with peroxide-like activity. This MNM can colorimetrically detect and remove catechol from water (Figure 19C) [208]. Catalytic MNMs functionalized with laccase (Lac) possess a unique three-dimensional hierarchical structure that ensures full exposure of the active sites. The Co_3_O_4_-C layer on the hemisphere’s surface catalyzes the degradation of H_2_O_2_, enabling the MNMs to move autonomously at velocities up to 171 µm/s even at low fuel concentrations (7 wt%). Thanks to the combination of autonomous motion and high peroxide activity, these MNMs can detect catechol with high sensitivity (detection limit: 2.24 µM) and degrade it rapidly via -OH generated through the Fenton-like reaction. The three-dimensional structure also allows for the integration of magnetically responsive materials, enabling the MNMs to be magnetically recycled after completing their task, thus preventing secondary contamination (Figure 19D) [209].

An example of engineered MNMs for environmental remediation is demonstrated by Guix et al. [210]. They utilized alkanethiol-functionalized, Au-sputtered, Pt-based microtubes to effectively capture and remove oil droplets, offering a potential solution to oil spills. However, the thiol groups, which have a strong affinity for both Pt and Au, can interfere with the Pt-catalyzed surface, leading to a significant reduction in the MNMs’ propulsion efficiency. To overcome this challenge, the researchers proposed a method in which MnO_2_ was deposited on the Pt surface. This coating effectively blocked the openings of the microtubes, preventing the chemisorption of thiol compounds onto the Pt surface, and thus preserving the catalytic properties of the MNMs [211]. As a result, the MNMs maintained their enhanced functionality even after being modified with alkanethiol, enabling their continued use in oil droplet capture and transport. In a similar vein, hydrophobic Mn- and Mg-based MNMs have also been successfully applied for oil droplet removal and capture [21,144].

The release of abiotic organic dyes from industries such as paint, textiles, and printing has become a significant environmental concern, necessitating effective methods for their degradation and recycling. To address this issue, Fe/Pt microtubules, fabricated using a rolled-up technique, have been utilized for Fenton-like degradation of dye molecules (Figure 19E) [6,101]. These chemically driven MNMs exhibit continuous movement for over 24 h and remain effective for multiple cleaning cycles over several weeks. In a similar vein, Wani et al. reported the dual functionality of MnO_2_ particles, which enable both catalytic degradation and bubble separation of rhodamine 6G and methylene blue dyes [212]. These particles are capable of self-degradation after several hours of use and, due to their low cost, large-scale availability, and transient nature, represent a viable solution for environmental applications. In addition to organic dyes, other persistent organic pollutants, commonly found in products like plastics, electronics, furniture, cosmetics, and personal care items, also present considerable environmental hazards. Reduced graphene oxide (rGOx), with its hydrophobic nature, adsorbs these pollutants through π–π stacking interactions [213]. SiO_2_ particles coated with rGOx nanosheets and Pt hemispheres are effective in removing polybrominated diphenyl ethers and triclosan from environmental samples (Figure 19E) [214]. To mitigate the bioaccumulation and persistence of these pollutants, Vilela et al. developed an MNM-in-sponge platform. This system combines the hydrophobic properties of sponges with the catalytic abilities of cobalt-ferrite (CFO) MNMs embedded in the sponge’s core, enabling in situ degradation of contaminants [215]. Operating with a low fuel concentration of 0.13%, the platform is not only reusable but also recyclable. Its bubble propulsion mechanism enhances fluid mixing and accelerates fluid exchange, ensuring efficient pollutant degradation over multiple cycles. To further improve the catalytic efficiency of MNMs for organic pollutant degradation, Gao et al. synthesized innovative MNMs that combine both selectivity and catalytic functionality. By using halloysite nanotubes (HNTs) as templates and modifying them with natural and nanoenzymes, they achieved enhanced catalytic performance. These MNMs demonstrated increased sensitivity for detecting minocycline (MC), a common water pollutant, and exhibited rapid degradation capabilities (Figure 19F) [216].

The accelerated industrialization of contemporary society has led to a notable increase in the accumulation of toxic heavy metals in both soil and aquatic ecosystems, highlighting the urgent need for effective regulatory measures and efficient remediation techniques. GOx nanosheets, which are rich in oxygen-containing functional groups, exhibit strong affinity for metal ions, facilitating the efficient adsorption of heavy metals from contaminated water. When employed as an outer layer on chemically driven MNMs, GOx significantly enhances the capture and removal of Pb^2+^, demonstrating a tenfold improvement in efficiency compared to non-motile counterparts [217]. An alternative promising method involves the functionalization of MNMs with metal-chelating agents. For example, Wang and collaborators developed meso-2,3-dimercaptosuccinic-acid-functionalized Au/Ti/Mg Janus MNMs, which effectively removed Zn^2+^, Cd^2+^, and Pb^2+^ from polluted water [218]. These water-powered MNMs operate without the need for external fuel and exhibit a lifespan of about three minutes. Within just two minutes, they achieve up to 100% metal removal efficiency. This approach is versatile and can be adapted with different chelating agents, allowing for selective removal of specific heavy metal contaminants.

While traditional catalytic MNMs powered by H_2_O_2_ can effectively degrade pollutants, they face challenges related to maintaining a consistent fuel supply and addressing the uneven distribution of the fuel, which limits their long-term efficacy in pollutant degradation. To overcome these obstacles, enzyme-driven MOF MNMs were developed [219]. Initially, pre-synthesized microporous MNMs were treated with ozone to create a mesoporous structure that facilitates the adsorption and incorporation of catalase. Catalase was then encapsulated within these mesopores, enabling the MNMs to exhibit autonomous motion. The hierarchical pore structure of these MNMs provides abundant space for contaminant adsorption, such as rhodamine B (Figure 19G). This innovation not only expands the range of enzymatic MNMs but also provides valuable insights into synergistic enzyme catalysis, advancing the potential of enzymatic MNMs in environmental remediation. Despite substantial progress in the synthesis of MNMs for environmental applications over the past decade, the development of pollutant-fueled MNMs capable of degrading multiple pollutants simultaneously remains a significant challenge. To address this, Gao et al. designed Fe_3_O_4_@SiO_2_ MNMs powered by laccase [220]. With the support of lipase, these MNMs utilized pollutants as their fuel source, thereby enhancing their ability to degrade multiple pollutants concurrently (Figure 19H).

In addition to facilitating chemical degradation, self-propelled MNMs can efficiently eliminate both organic and inorganic pollutants from contaminated water via adsorption or selective binding mechanisms [221,222,223]. One of the most critical environmental challenges today is the release of CO_2_ into the atmosphere, which significantly contributes to global climate change. Traditional methods for capturing and storing CO_2_ are not only energy-demanding and expensive but also involve the use of toxic chemicals that may cause secondary environmental harm [224]. A promising solution to this issue is biocatalytic CO_2_ sequestration, as demonstrated by Uygun et al., who employed carbonic anhydrase-immobilized microtubules [225]. These self-moving MNMs catalyze the hydration of CO_2_, converting it into CaCO_3_, thus presenting a potential approach for developing mobile microsystems that could enhance the efficiency of CO_2_ capture and storage.
Figure 19Chemically driven MNMs for environmental remediation. (**A**) Motion of Pt/Au nanorods in the presence of 100 × 10^−6^ m metal–nitrate salt solution of different metals, in 5% H_2_O_2_. The speed enhancement in the presence of Ag^+^ ions provides a route to determine the aquatic Ag^+^ toxicity. Reproduced from Ref. [201]. Copyright 2010, American Chemical Society. (**B**) Top: Catalase-immobilized tubular MNMs; bottom: the effect of pollutants on the locomotion speed due to the inhibition of catalase. Reproduced from Ref. [203]. Copyright 2013, American Chemical Society. (**C**) A 3D hierarchical LDH-based Janus microactuator for detection and degradation of catechol. Reproduced from Ref. [208]. Copyright 2023, Elsevier B.V. All rights reserved. (**D**) A 3D hierarchical HRP-MIL-100 (Fe)@ TiO_2_@ Fe_3_O_4_ Janus magnetic micromotor as a smart active platform for detection and degradation of hydroquinone. Reproduced from Ref. [209]. Copyright 2022, American Chemical Society. (**E**) Conversion of an organic dye into byproducts due to a Fenton-like reaction by Fe/Pt tubular MNMs. Reproduced from Ref. [6]. Copyright 2013, American Chemical Society. (**F**) MNMs modified with a mixture of natural enzymes and nanoenzymes were used for pollutant degradation. Reproduced from Ref. [216]. Copyright 2023, Royal Society of Chemistry. (**G**) Enzyme-powered porous micromotors built from a hierarchical micro- and mesoporous UiO-type metal–organic framework. Reproduced from Ref. [219]. Copyright 2020, American Chemical Society. (**H**) Contaminant-fueled laccase-powered Fe_3_O_4_@SiO_2_ nanomotors for synergistical degradation of multiple pollutants. Reproduced from Ref. [220]. Copyright 2022, Elsevier Ltd. All rights reserved.
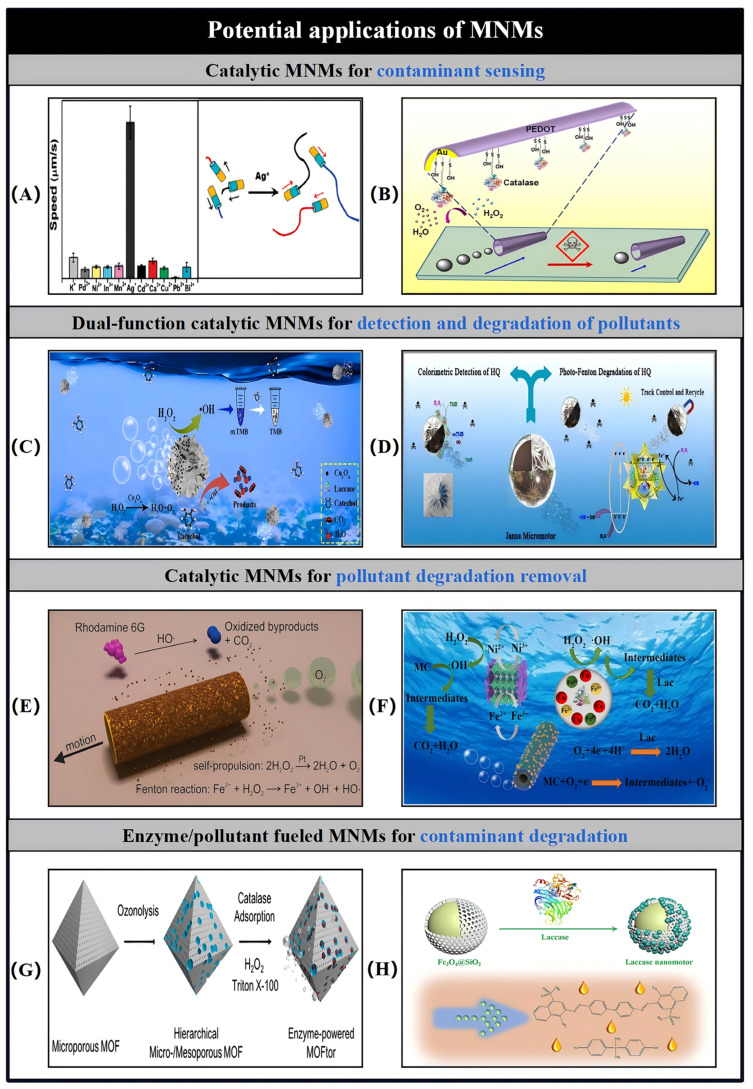


## 8. Biosafety and Biocompatibility

In the previous section, we examined the various applications of chemically driven MNMs, emphasizing their promising role in the biomedical sector. Despite notable advancements in the use of chemically powered MNMs within living systems, their successful application in vivo still demands further investigation into the intricacies of the biological environment. A consensus among researchers underscores the importance of designing chemically driven MNMs with safety as a primary consideration to ensure their effective functioning within the body. This involves not only confirming the biosafety and biodegradability of the MNM structure but also evaluating the safety of the chemical fuels utilized [226]. In conclusion, for the successful deployment of chemically driven MNMs in biological settings, it is essential to address critical factors such as biocompatibility, potential immune responses, organ toxicity, degradability, and the practical challenges associated with removing the MNMs from the body [227].

### 8.1. Component Characteristics

To address biosafety concerns related to MNMs, researchers have investigated the use of reactive metals and porous SiO_2_ as foundational materials for constructing MNMs [196,228]. These MNMs are engineered to degrade either autonomously or under specific environmental conditions, such as changes in pH (Figure 20A) or in the presence of particular enzymes (Figure 20B), once their intended function is completed [229,230]. To assess their compatibility with biological systems, several in vitro tests, including cytotoxicity and hemolysis assays, were performed. For example, Pumera et al. found that elevated concentrations of Mg/Pt Janus MNMs caused negative effects on a variety of human cell lines, such as A549 (lung cancer), MCF-7 (breast cancer), HEK293 (embryonic kidney), HepG2 (liver cancer), and HeLa (cervical cancer) cells [231]. These findings suggest that Mg/Pt MNMs may not be entirely suitable for biological applications. Reactive metals, however, provide flexibility in MNMs design, enabling the creation of structures like spherical, tubular, and Janus MNMs. Expanding on this approach, Wang et al. introduced transient, self-destructive MNMs, including Mg/ZnO, Mg/Si, and Zn/Fe Janus MNMs, as well as single-component Zn MNMs (Figure 20C) [232]. These MNMs are specifically designed to dissolve in biological media after completing their tasks, thereby reducing the buildup of toxic residues. Importantly, the Mg^2+^ and Zn^2+^ ions released during the degradation of these MNMs are considered non-toxic and may even exert beneficial effects on tissue cells [233,234].

### 8.2. Chemical Fuel

To fully investigate the wide-ranging potential of MNMs in biomedical applications, it is crucial to consider not only the biosafety of the materials used in their construction but also the safety of the chemical fuels that power their movement. Among these fuels, H_2_O_2_ is frequently utilized because it produces gas bubbles through catalytic decomposition, a process facilitated by materials like Pt and MnO_2_ [235,236]. Despite its effectiveness, the inherent toxicity of H_2_O_2_ significantly limits the use of MNMs in biological systems. Efforts to reduce the concentration of H_2_O_2_ have been explored as a means of minimizing its harmful effects; however, this approach often results in a decrease in the movement speed and overall efficiency of the MNMs [181]. As a result, there has been increasing interest in finding alternative, safer fuels, leading to the investigation of enzyme-based propulsion systems [237]. Enzyme-powered MNMs, utilizing biocompatible fuels such as urea and glucose, present a promising solution, offering the potential for self-powered MNMs with improved biosafety profiles [238,239].

Urea, a natural byproduct of protein metabolism, is commonly present in living organisms and is well-documented in scientific research [240]. When broken down by urease enzymes, urea releases CO_2_ and NH_3_ [241], which not only generate the mechanical force necessary for the movement of MNMs but also contribute to creating a less toxic environment. These gases are utilized by MNMs to facilitate the transport of particles through the bladder, addressing various bladder-related issues. The movement of MNMs was found to increase with higher concentrations of urea, eventually reaching a point of saturation. In addition, biocompatibility tests conducted on MNMs with human bladder cells confirmed the safety and practicality of these systems for in vivo applications [242]. Furthermore, MNMs can also extract urea from gastric juices, utilizing it as a bioavailable energy source for propulsion. A histological examination of the gastrointestinal tissues of mice was performed three days after the oral administration of MNMs. The findings indicated that the MNMs were eliminated from the body without causing any harmful effects to the surrounding tissues. To further assess their clinical potential, a thorough biodegradability analysis was carried out. The results indicated that MNMs degrade swiftly under oxidative conditions (Figure 20D) [243].

Glucose serves as a vital energy source within the circulatory systems of living organisms, where it undergoes enzymatic breakdown by glucose oxidase, producing gluconic acid and H_2_O_2_ [244]. This metabolic process positions glucose as a promising biocompatible fuel, capable of powering the autonomous movement of MNMs. Battaglia et al. took advantage of the glucose concentration gradient across the blood–brain barrier to guide the transport of glucosidase-based MNMs, illustrating their chemotactic movement [245]. When compared to non-template systems, these MNMs demonstrated a fourfold enhancement in brain penetration, allowing them to move effectively under physiological glucose conditions. Additionally, substances like gastric acid, which are naturally acidic within organisms, can also serve as chemical energy sources for driving MNMs. When Mg-loaded MNMs react with gastric acid, H_2_ gas bubbles are produced, propelling the MNMs and facilitating their role in targeted drug delivery for the treatment of gastric ulcers in mice [246].
Figure 20Safety and biocompatibility of chemically powered MNMs. (**A**) Toxicity evaluation of Mg MNMs. Scale bars, 100 mm. Reproduced from Ref. [229]. Copyright 2017, Wiley-VCH GmbH. (**B**) Hybrid biodegradable MNMs through compartmentalized synthesis. Reproduced from Ref. [230]. Copyright 2020, American Chemical Society. (**C**) Design of Mg/ZnO Janus MNMs and Zn/Fe Janus MNMs. Reproduced from Ref. [232]. Copyright 2016, American Chemical Society. (**D**) Histological analysis with H&E staining of stomach (first row), duodenum (second row), ileum (third row), and colon (last row) at 3 days post-injection of PBS and MNMs (scale bar = 200 µm). Reproduced from Ref. [243]. Copyright 2022, KeAi Communications Co.
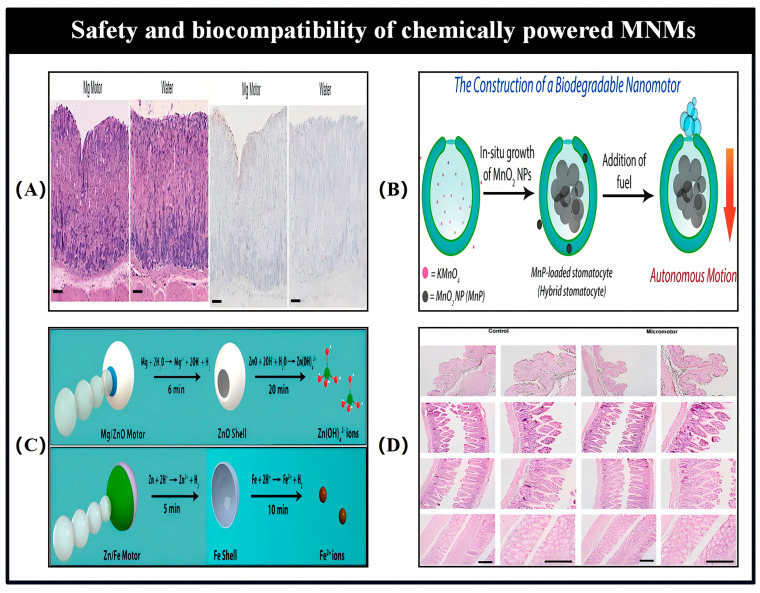


### 8.3. Biological Environment

The microenvironment that supports cell survival is primarily shaped by the extracellular matrix, its humoral elements, and the surrounding biological conditions. When used in vivo, MNMs can significantly modify this microenvironment, potentially affecting cellular functions. As a result, researchers have increasingly focused on understanding how MNMs within biological environments impact the physiological behavior of normal cells or interfere with pathological processes. The actions of immune cells, especially in the context of tissue repair and regeneration within an inflammatory environment, are of particular importance. Therefore, it is critical to examine how MNMs may influence various immune cell phenotypes, including those promoting inflammation, tissue repair, and fibrosis. Moreover, exploring how MNMs can be effectively utilized for immunomodulation represents a significant area of research [247,248]. Another important factor is the interaction between MNMs and blood components, such as platelets and red blood cells. Investigating whether MNMs could induce blood clot formation and the mechanisms that regulate clot resolution is crucial. Additionally, understanding the potential interactions between MNMs and beneficial microorganisms in the body is essential. In summary, the biosafety of MNMs is intricately linked to maintaining homeostasis, as well as facilitating tissue regeneration and repair within living organisms [227,249].

Chemically powered MNMs have been found to enhance angiogenesis while also demonstrating bactericidal effects. These MNMs promote the proliferation and migration of endothelial cells, leading to a marked increase in vessel density. During the recovery process, they further stimulate endothelial cell movement and protein deposition. Additionally, NO released in the later stages aids in the healing of skin and prevents the infiltration of inflammatory cells into healthy tissues. These observations underscore the favorable biocompatibility of this particular class of MNMs [250]. Furthermore, the application of self-driven forces aligned perpendicular to the flow field has been shown to enhance the ability of MNMs, loaded with nanoparticle-based drugs, to penetrate blood vessels. In vivo studies have demonstrated that these MNMs accumulate more efficiently at tumor sites, suggesting that their self-propulsive movement aids in their penetration through intercellular spaces between endothelial cells without significantly affecting major organs or altering cell morphology [251].

MNMs containing PMA-TPP/PTX have been found to improve endothelial cell function while reducing oxidative stress. The gas produced by these MNMs supports the survival and proliferation of healthy cells, aiding in the restoration of endothelial barrier integrity. Histological analysis indicates that organs maintain their normal morphology with no significant tissue damage [252]. Urease-powered MNMs enable deeper penetration of chemotherapeutic agents, and when combined with photothermal treatment, they demonstrate enhanced anticancer activity. These MNMs degrade in physiological conditions without causing notable harm to the skin or surrounding tissues [253]. Similarly, NO-powered MNMs, when applied to skin wounds with compromised function due to bacterial infection, facilitate bacterial clearance and accelerate the healing process. Tissue examination conducted 12 days after treatment shows organized collagen deposition, largely preserved tissue structure, and no significant abnormalities or damage [254].

## 9. Conclusions and Outlook

Catalytic MNMs offer distinct advantages, including flexible and variable shapes, superior motion characteristics, and strong mechanical properties, allowing them to adapt effectively to complex motion environments and accomplish intricate tasks upon reaching target sites. However, several challenges limit their practical application: 

(i) drive efficiency and environmental adaptability: the drive efficiency of catalytic MNMs heavily depends on specific fuels that are often scarce, easily consumed, and potentially toxic in biological or natural environments. High-viscosity media, uneven surfaces, and low fuel concentrations further reduce their propulsion efficiency due to viscous drag and bubble formation. Future research should focus on developing biofuel-based MNMs that utilize environmental resources (glucose, O_2_ in organisms, or pollutants in aquatic environments) and integrating stimulus-responsive materials to enable multiple driving mechanisms, enhancing adaptability in complex environments. 

(ii) Control of precision and group synergy: catalytic MNMs usually rely on spontaneous chemical reaction drive, and their motion paths are susceptible to external disturbances such as hydrodynamic effects, environmental noise, and uneven fuel distribution. When they are used to perform complex tasks such as targeted drug delivery and pollutant removal, the motion trajectories of catalytic MNMs need to be accurately controlled, however, their random motion patterns can limit the effectiveness of the actual tasks. Meanwhile, the insufficient regulation of group synergistic behavior is another factor that affects the effectiveness of its application. When using group-catalyzed MNMs to perform complex tasks, individuals within the group may be out of control or disconnected, resulting in lower efficiency. In addition, random motion and noise interference is another important factor that affects the control accuracy of catalytic MNMs. Since the catalytic MNMs are too small in size, their motion behaviors are susceptible to thermal Brownian motion, fluid eddy currents, or local chemical gradient variations, leading to a sharp increase in trajectory randomness and a decrease in task accuracy. Therefore, there is a need to introduce intelligent control techniques (control strategies based on artificial intelligence or machine learning) into catalytic MNM systems to enhance the real-time control of individual and group behaviors by means of heuristic rules such as velocity matching, coherence, and cohesion. In addition, by further designing responsive materials or structures, MNMs capable of environmental sensing are developed to enable adaptive regulation based on external signals (chemical gradients or physical fields). 

(iii) Stability and long-time operation capability: during chemically driven processes, the catalyst surface may combine with impurities or by-products in the fuel to form a deposition layer that is difficult to decompose, leading to a loss of catalytic activity. Also, during long-term operation, the catalyst may degrade due to thermal effects, chemical corrosion, or mechanical damage, resulting in the efficiency of catalytic MNMs being reduced and them being unable to maintain steady motion. At the same time, catalytic MNMs consisting of metals/oxides are susceptible to destruction at high reaction rates or in corrosive environments (acidic, alkaline or oxidizing media). During long-term operation, the structural integrity of the material may be affected by microscopic cracks or surface wear, leading to loss of function. In addition, the accumulation of reaction products may hinder the surface-active sites of the catalysts as well as the generation of harmful by-products that can interfere with the proper functioning of the system. In order to enable catalytic MNMs to perform high-precision tasks with strong stability and long-time operation capability, firstly, new high-stability catalysts, such as nanoenzymes and MOFs, should be used to construct catalytic MNMs. Secondly, their surfaces should be decorated using self-healing materials or coatings with corrosion resistance to enhance their long-time operation capability while protecting the catalysts from toxicity or contamination. 

(iv) Biocompatibility and safety: the fuels used in catalytic MNMs may be toxic to biological tissues at high concentrations, leading to cell damage or death. Their own material composition may also produce toxic by-products due to the release of ions or degradation, further threatening biosafety. Furthermore, the body’s immune system may recognize catalytic MNMs as foreign substances, activating macrophages or inducing an inflammatory response, which may affect tissue function and reduce therapeutic efficacy. In addition, MNMs manufactured using non-degradable or partially degradable materials may remain in the body after the task is accomplished, and their long-term accumulation may lead to chronic inflammation or other health risks. Therefore, efforts should be made to develop fully degradable MNMs based on biocompatible materials (natural polymers or biocompatible metals) to ensure that they leave fewer residues in the body after completion of the task. And the invisibility and biosafety of MNMs should be enhanced by surface modification techniques (adding biomolecular coatings). 

(v) Preparation technology and scale-up production: the preparation process of high-performance MNMs involves complex chemical reactions, material modification and functionalization steps, and the high cost of some metal materials, thus leading to a complex and costly preparation process of catalytic MNMs. Meanwhile, the existing preparation processes are limited to small-scale laboratory processes and cannot be converted into industrial production processes. In addition, the inhomogeneity of the material distribution, structural dimensions or functionalization process during batch preparation leads to significant differences in individual performance, reducing the controllability and predictability of the population behavior. Subsequent research should be devoted to the development of simplified and low-cost preparation processes, such as template and self-assembly methods, and the synergistic use of advanced fabrication technologies (3D printing) to achieve high-precision preparation. In addition, modular design is adopted to facilitate rapid assembly and industrial production of materials and structures. 

(vi) Complexity of application scenarios and multifunctional integration: complex application scenarios (complex liquid environments in vivo or heavily polluted water bodies) may have multiple interfering factors that affect the movement efficiency and functionality of MNMs. Moreover, single-function MNMs can no longer meet the diversified task requirements. Therefore, MNMs should be built with multifunctionality by integrating actuation, sensing, response, and functional outputs in the same platform, and the material and structure of MNMs should be optimized for specific application scenarios (biomimetic design) to enhance adaptability.

## Figures and Tables

**Figure 1 nanomaterials-15-00013-f001:**
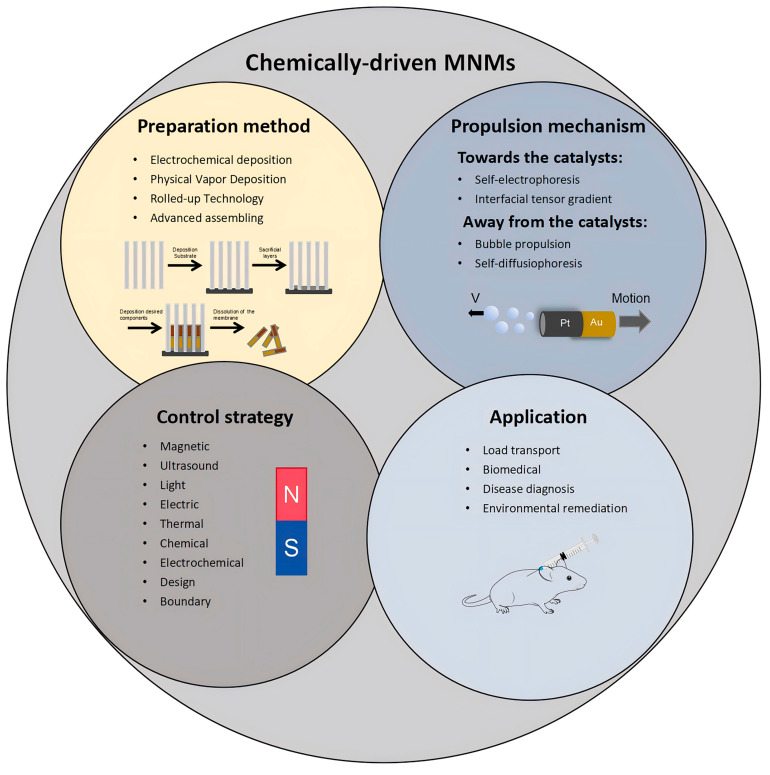
A diagrammatic overview illustrating the fabrication process, propulsion mechanism, control strategies, and potential applications of catalytic MNMs.

**Figure 2 nanomaterials-15-00013-f002:**
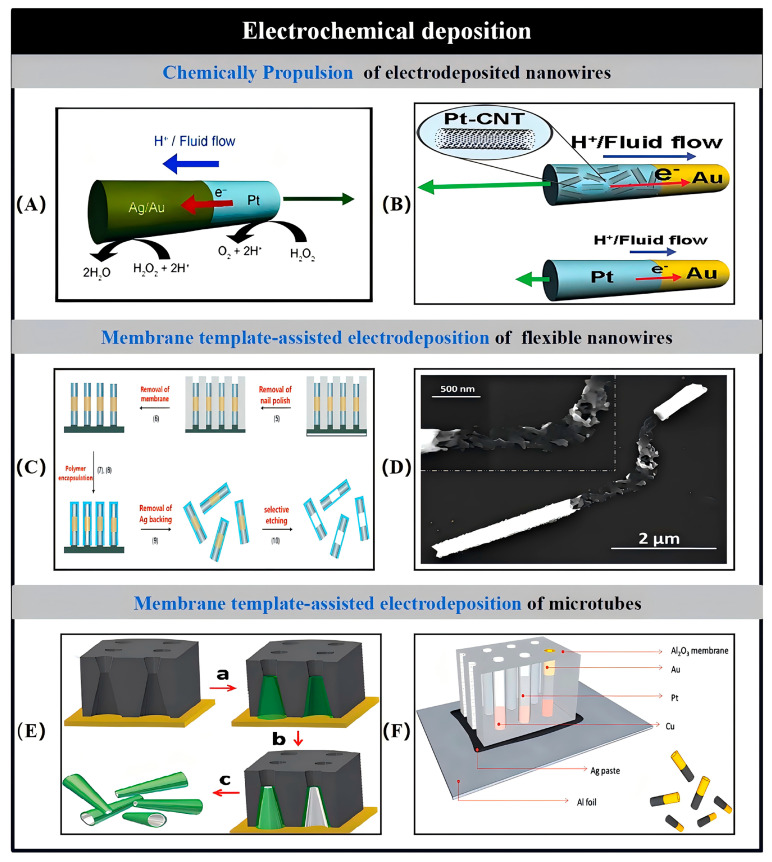
Examples of MNMs grown by template-assisted electrochemical deposition. (**A**,**B**) Schematic representations of the self-electrophoresis mechanism of Ag-Au/Pt and Au/Pt-CNT nanowire MNMs in H_2_O_2_, respectively. Reproduced from Refs. [36,37]. Copyright 2008, Wiley-VCH and 2008, the American Chemical Society, respectively. (**C**) Preparation procedure of flexible metallic nanowires with polyelectrolyte hinges after membrane template electrodeposition. (5) removal of nail polish, (6) dissolution of the membrane, (7) deposition of mercaptoethanesulphonic acid, (8) layer-by-layer deposition of poly(sodium-4-styrenesulphonate) (PSS) and poly(diallyldimethyl ammonium chloride)(PDMAC), (9) removal of Ag by nitric acid, and (10) etching of the Au segment using KI/I_2_ solution. Reproduced from Ref. [39]. Copyright, 2007 Nature Publishing Group. (**D**) SEM image of Au/Ag/Ni MNMs with flexible central silver segment. Reproduced from Ref. [41]. Copyright, 2010 the American Chemical Society. (**E**) Polycarbonate membrane-assisted preparation of conical PANI/Pt microtubes. (a) deposition of the polyaniline (PANI) microtube, (b) deposition of the Pt microtube, and (c) dissolution of the membrane and release of the bilayer microtubes. Reproduced from Ref. [38]. Copyright 2011, American Chemical Society. (**F**) Anodized aluminum oxide (AAO) membrane-assisted preparation of segmented microtubes. Reproduced from Ref. [40]. Copyright 2013, Royal Society of Chemistry.

**Figure 4 nanomaterials-15-00013-f004:**
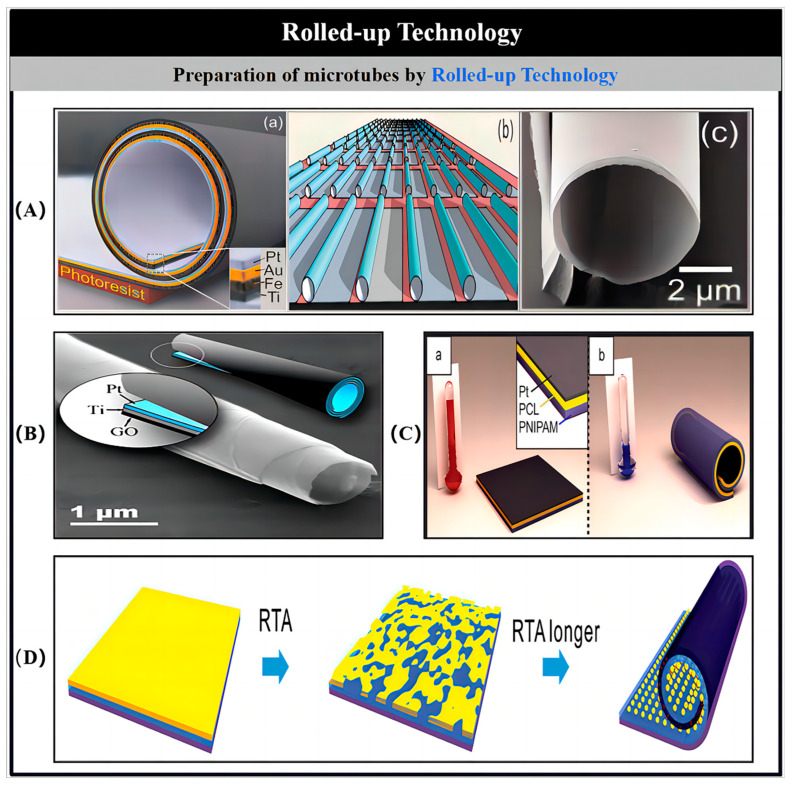
MNMs prepared by rolled-up technology. (**A**) Rolling-up of nanomembranes patterned with photoresist: (**a**,**b**) schematic illustration of a rolled-up microtube comprising Pt/Au/Fe/Ti multilayers on a sacrificial photoresist layer and an array of rolled-up microtubes, respectively; (**c**) SEM image of a rolled-up microtube. Reproduced from Refs. [52,53]. Copyright 2009, Wiley-VCH and 2010, Wiley-VCH, respectively. (**B**) Rolled-up microtubes with GO as an external layer. Reproduced from Ref. [54]. Copyright 2012, the American Chemical Society. (**C**) Reversible rolling and unrolling of thermo-responsive polymeric Pt microtubes. (**a**) shows the 3 layers of poly(NIPAM-BA), PCL (polycaprolactone), and Pt. (**b**) Representation reversible rolling–unrolling of polymer-Pt films. Reproduced from Ref. [55]. Copyright 2014, Wiley-VCH. (**D**) Particle-aided rolling process of nanomembrane upon a thermal dewetting treatment. Reproduced from Ref. [56]. Copyright 2013, Wiley-VCH.

**Figure 5 nanomaterials-15-00013-f005:**
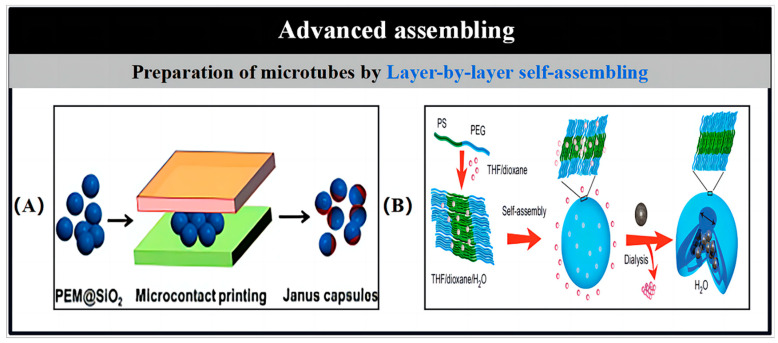
Fabrication of MNMs by assembly of materials. (**A**) Synthesis process of Pt-NP-functionalized Janus capsule motors. Reproduced from Ref. [61]. Copyright 2012, the American Chemical Society. (**B**) Selective and controlled encapsulation of Pt NPs inside artificial stomatocytes during shape transformation. Reproduced from Ref. [62]. Copyright 2012, Macmillan Publishers Limited.

**Figure 6 nanomaterials-15-00013-f006:**
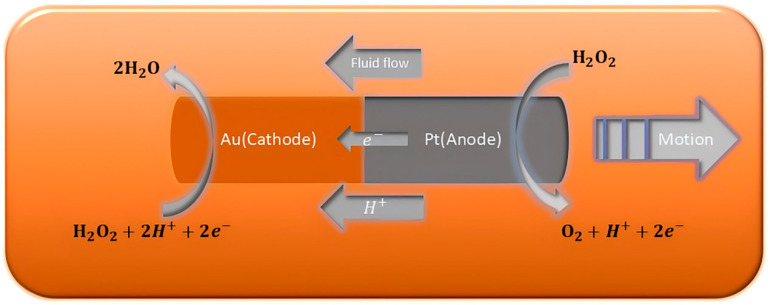
A schematic representation illustrates the dielectrophoresis mechanism responsible for the propulsion of Au/Pt MNMs in the presence of H_2_O_2_. This mechanism involves an internal electron flow from one end of the MNMs to the other, accompanied by the migration of protons within the double layer surrounding the MNMs. The interaction between these two processes generates a propulsion force, enabling the autonomous motion of the MNMs.

**Figure 7 nanomaterials-15-00013-f007:**
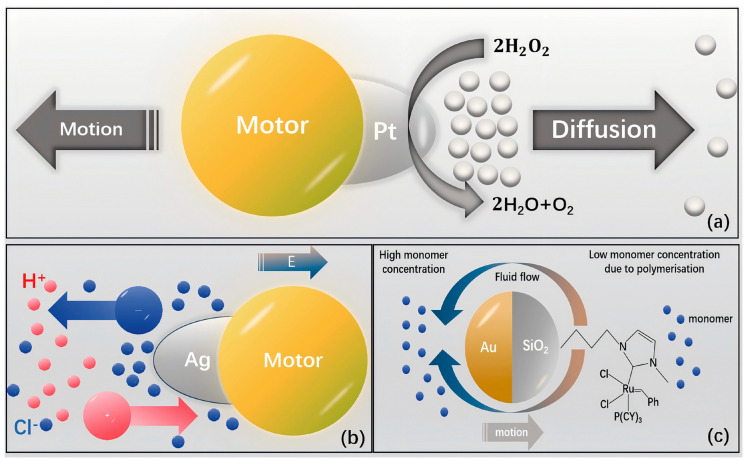
Schematic representation of MNMs moving under the diffusiophoresis propulsion mechanism. (**a**) Self-diffusiophoresis. (**b**) Ionic self-diffusiophoresis. (**c**) Non-ionic diffusiophoresis.

**Figure 8 nanomaterials-15-00013-f008:**
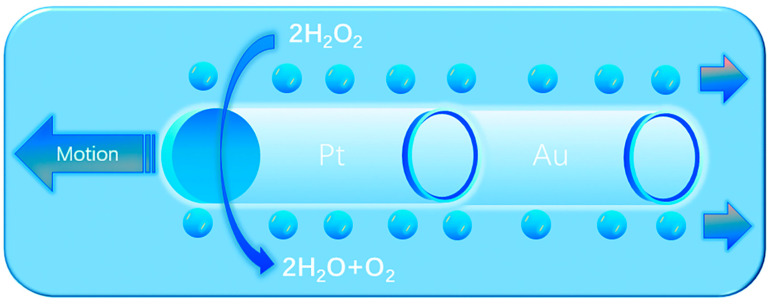
Schematic diagram of MNMs propelled under the interfacial tension propulsion mechanism.

**Figure 9 nanomaterials-15-00013-f009:**
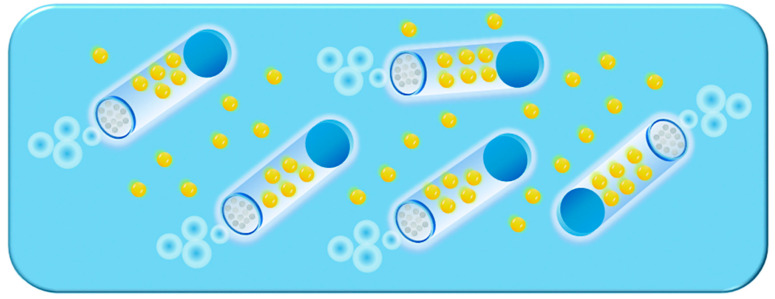
Schematic representation of MNMs moving under the bubble propulsion mechanism. The inner surface is constructed from a Pt catalyst. During the H_2_O_2_ decomposition, O_2_ bubbles are generated and released from the wider end of the motor resulting in the propulsion of the motor away from the bubbles and the catalyst.

**Figure 10 nanomaterials-15-00013-f010:**
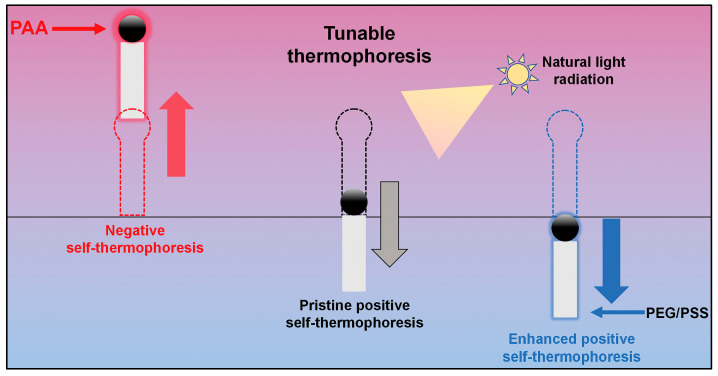
Tunable Self-Thermophoretic MNMs.

**Figure 11 nanomaterials-15-00013-f011:**
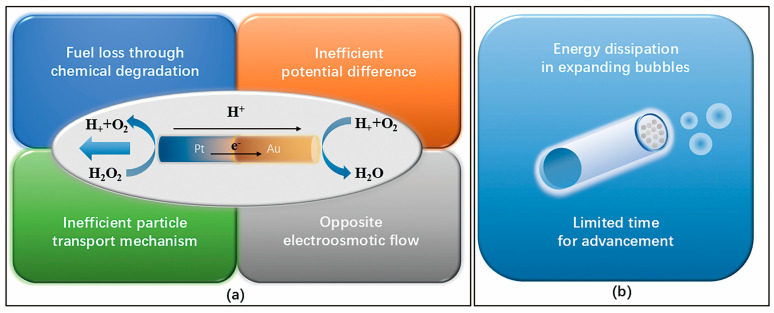
Reasons for the inefficiency of MNMs during Self-electrophoresis mechanism and bubble recoil mechanism. (**a**) In the self-electrophoresis propulsion mechanism, the four energy loss stages of MNMs during the conversion process. (**b**) In the bubble recoil propulsion mechanism, MNMs have two stages of energy loss during the conversion process.

**Figure 14 nanomaterials-15-00013-f014:**
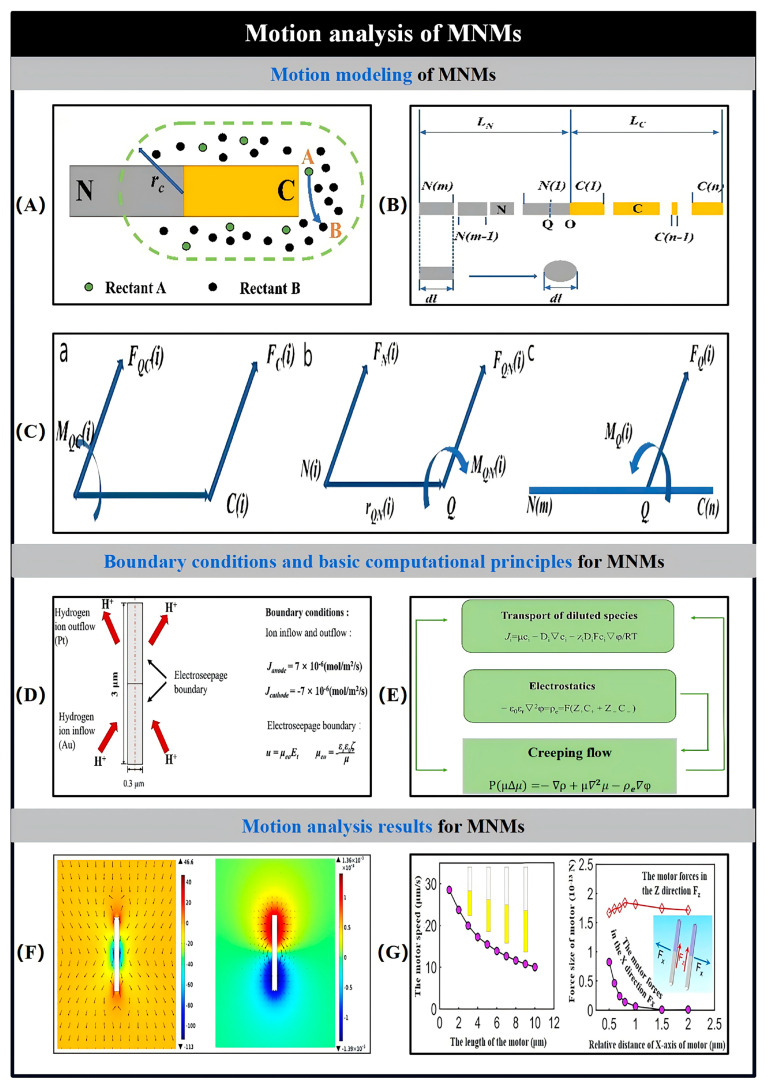
MNM Motion Simulation Analysis. (**A**–**C**) Motion modeling of MNMs. (**A**) The chemical reaction area at the catalytic end of the nanowire MNMs. (**B**) Schematic diagram of finite section division. C(1)–C(n) is the catalytic finite segment, n is the number of catalytic segments, N(1)–N(m) is the non-catalytic segment, m represents the number of non-catalytic segments, O represents the catalytic segment and non-catalytic segment junction point, Q represents the center of mass of the nanowire MNMs. (**C**) Force analysis of MNMs. (**a**): The force acting on the catalytic part; (**b**): The force acting on the non-catalytic part; (**c**): The force equivalent to the center of mass of the nanowire MNMs. Reproduced from Ref. [152] Copyright 2022, Wiley-VCH. (**D**,**E**) Boundary conditions and basic computational principles for MNMs. (**D**) Two-dimensional axisymmetric double metal bar model boundary conditions. (**E**) Basic calculation principle of the model. Reproduced from Ref. [152]. Copyright 2022, Wiley-VCH. (**F**,**G**) Motion analysis results for MNMs. (**F**) Flow field and electric field distribution around the MNMs. Reproduced from Ref. [157]. Copyright 2013, American Chemical Society. (**G**) The relationship between the speed and length of a single MNM, and the force on the MNMs when two MNMs are opposite each other. Reproduced from Ref. [152]. Copyright 2022, Wiley-VCH.

**Figure 15 nanomaterials-15-00013-f015:**
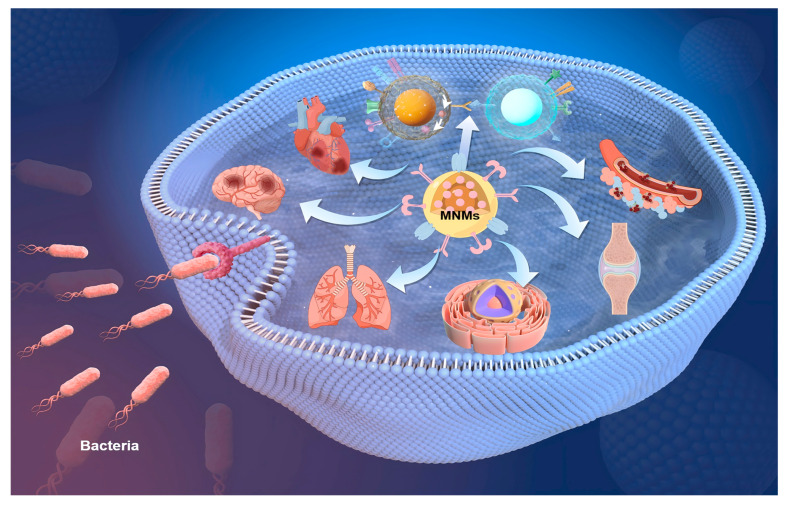
Application of MNMs in the direction of biomedicine.

**Figure 18 nanomaterials-15-00013-f018:**
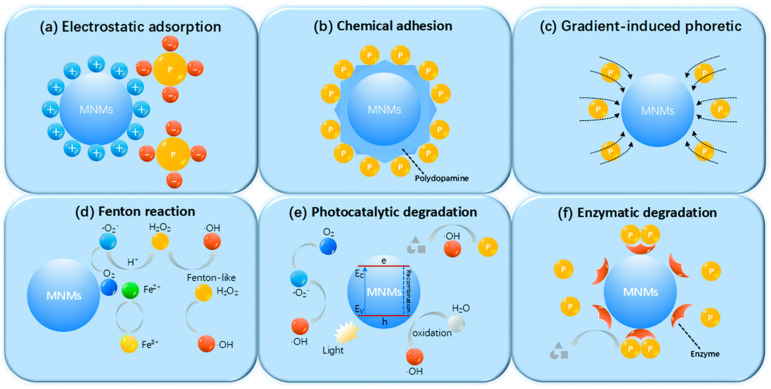
Mechanisms of pollutant removal and degradation by MNMs.

**Table 1 nanomaterials-15-00013-t001:** Main MNM fabrication methods.

Fabrication	Technique	Structure	Advantages	Limitations
Electrochemical	Membrane template-assisted electrodeposition, asymmetric bipolar electrodeposition	Nanowires, nanorods, microtubes	Low cost, prepared from inorganic or organic materials	Large-scale production and MNM geometries limited by membrane features (pore size, shape)
Physical vapor deposition	Sputtering, evaporation, atomic layer deposition, glancing angle deposition	Anisotropic structures, coatings for Janus structures	Highly controllable and reusable	Expensive equipment
Rolled-up	Self-scrolling technique	Vesicles, Janus, polymersomes	Flexible structural design and control, highly versatile, high scalability	Complex processes, stress distribution and shape of the material are difficult to control
Self-assembly	Layer-by-layer assembly, macromolecular assembly, shape transformation	Vesicles, Janus, polymersomes	Simple equipment, low-cost, sustainability, versatility, bio- inspired and biodegradable materials preparation	Mostly fuel-driven

**Table 2 nanomaterials-15-00013-t002:** Comparison of different driving methods for MNMs.

Driving Method		Sports Performance	Span	Biological Adaptation	Biosafety	Limitations
Physical	Magnetic	Accurate 3D Motion	MNMs are capable of sustained motion under the action of an applied field, with better sustained performance	Good tissue penetration and applicability	Better biocompatibility	Electromagnetic drive with low energy efficiency and limited operating space
	Light	Start–stop control and directional motion (fast motion)		Poor tissue penetration, only suitable for superficial tissues	UV light is harmful to the body	High-power light sources are required in liquids with limited light focus size and range of motion
	Ultrasound	Fast response time		Good tissue penetration and applicability	Comparatively safe	Lack of operational precision
	Electrical	Convergent movement		Weak biopenetration, requiring strong electric field strengths	Strong electric fields can affect the human body	Low range of motion
Biologically		Relying on the motor properties of individual organisms	Related to biological cell activity	Partially applicable	Biocompatible	Biological cells/bacteria need specific nutrient environments to survive
Chemically		High speed of movement	Poor sustained performance due to gradual depletion of chemical fuels	Partially applicable	Common chemical fuels are harmful; enzyme-initiated biocatalytic reactions are biocompatible	Speed of movement depends on fuel concentration; poor persistence; direction of movement uncontrollable

**Table 3 nanomaterials-15-00013-t003:** Main control methods of MNMs.

Types of Control	Function	Instructions
Magnetic field	Magnetic guidance	To control the direction of motion by orienting or changing the strength of a magnetic field
Ultrasonic	For trend	The MNMs converge in the direction of high peroxide concentration
Electric	For assembly	Dynamic assembly under rotational torque
Optical	Reaction switches and speed controls	Different light sources are used for motion control and switching control
Thermal control	Real-time speed control	The speed is adjusted by means of local heat pulse
Chemical control	Speed control	Control of movement speed by tuning the local fuel level
Electrochemical	Reaction switch	An external electric field is applied to control the start and stop of motion
Design control	For more complex movement patterns	Giving MNMs an asymmetric geometry making them have a rotational component
Boundary control	Elimination of the need for external energy sources	Using the interaction of MNMs with the surrounding wall to plan their kinematic behavior

**Table 4 nanomaterials-15-00013-t004:** Self-driving MNM speed expression.

Drive Method	Velocity Expression Equation	Instructions
Self-electrophoretic	Smoluchowsky equation: $V=\frac{\xi \theta }{\mu }E$	ξ: the MNMs’ zeta potential; θ: solution permittivity; μ: hydrodynamic viscosity;
Non-electrolyte diffusion	Diffusion swimming model: $U=\frac{kT}{\mu }KL\Delta C$	k: Boltzmann’s constant; T: solution temperature; K: the Gibbs adsorption length; L: the interaction length of the solute and MNM surface; ΔC: the solute concentration gradient.
Electrolyte diffusion	Electrophoresis+chemical swimming: $\begin{array}{c} V=(\frac{dIn\left(C\right)}{dx})\left[\left(\frac{D_{c}-D_{a}}{D_{c}+D_{a}}\right)\left(\frac{kT}{e}\right)\frac{\varepsilon \left(\xi_{p}-\xi_{s}\right)}{\mu }\right]\\ +\left(\frac{dIn\left(C\right)}{dx}\right)\{(\frac{2\varepsilon k^{2}T^{2}}{\mu e^{2}})\\ \left[In(1-(tan\frac{e\xi_{s}}{4kT}{)}^{2})-In(1-(tan\frac{e\xi_{s}}{4kT}{)}^{2})\right]\} \end{array}$	$dIn\left(C\right)/dx$: the electrolyte concentration gradient; $D_{a},D_{c}$: the diffusion rates of cations/anions; k: the temperature of the T-type solution; e: the elementary charge; ε: the dielectric constant; $\xi_{p},\;\xi_{s}$: the zeta potentials of both the MNMs and the substrate.
Bubble propulsion	V=0; V=C (C is a constant). When the bubbles are discharged irregularly, the resultant force F_d_ of MNMs is 0, and no movement can be produced; when the bubbles are discharged in a certain direction, the resultant force F_d_ > 0 of MNMs moves in a certain direction.	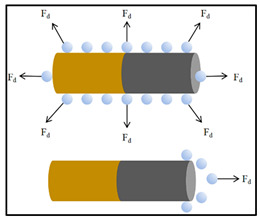
Interfacial tension gradient	$v=\frac{SR^{2}\gamma }{\mu DL}$	where S represents the O_2_ generation rate, R denotes the radius of the MNMs, γ is the surface tension of the solution, μ refers to the viscosity of the solution, D is the diffusion coefficient, L indicates the length of the MNMs, and k is a constant.

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
