# Peer review of "A Lifetime of Catalytic Micro-/Nanomotors"

_nanomaterials, 2024, doi:10.3390/nano15010013_

Round 1

Reviewer 1 Report

Comments and Suggestions for Authors

The article is well-written and organized, however the reviewer has some comments. There exist already many types of reviews on the topic in different journals thus the novelty and impact of the review is reduced. Many parts are written for specific geometries, the motion control for example is mostly focused on nanowires/nanorods/nanotubes, i.e. 1D geometry. Something that I liked is the addition of simulating the motion, introducing an overview on the computational characterization. However, I miss a part on the experimental motion characterization a discussion on the low-Reynolds number and the usually used mean square displacement (MSD) to characterize Catalytic Micro-/Nanomotors and what are the figures of merits for these systems. Please revise text-size for Figure2 and the quality of the image 3 as some look distorted and format correctly equation 6-18 as it is not in the right place.

Author Response

  1. Many parts are written for specific geometries, the motion control for example is mostly focused on nanowires/nanorods/nanotubes, i.e. 1D geometry.

In section 2.1:

To overcome the limitation of the complex fluid environment on the kinematic behaviour of catalytic MNMs, non-regularly shaped catalytic MNMs have been successively created. For example, Liu et al prepared Pd nanosprings by electrochemical deposition supported by anodic Al2O3 membrane using nanochannels42. The hydroxyl-terminated surface of the nanochannel can selectively absorb H+ to form a compact layer under suitable acid-base conditions. In the presence of an effective potential and an electroplating solution consisting of PdCl2, CuCl2 and HCl, the hydroxyl-terminated surfaces of the Al2O3 nanochannels and the localised precipitation of hydrogen contribute to the growth of Pd atoms at the outer sites of the Al2O3 nanochannels. Under the effect of helical dislocation, Pd is automatically wound onto Cu nanorods, and Pd nanosprings can be obtained after selective removal of Cu. When template-assisted electrodeposition is used to prepare helical MNMs, their diameters and lengths can be adjusted by adjusting the nanopore diameters and the ion concentration in the plating solution. In addition, magnetic materials can be deposited on the surface of the helical structures to achieve precise magnetic navigation43.

The template employed for electrochemical deposition can take the form of a porous membrane, with the shape of the template being selected according to the structure being prepared. In order to simplify the electrodeposition process, Manesh et al.44 used a simplified template-assisted layer to prepare catalytic conical MNMs. By sequentially depositing Pt and Au on etched Ag wires, the templates were then diced and dissolved. This approach enables precise control of the parameters of the catalytic MNMs, thereby enhancing their properties. The catalytic MNMs are driven by an internally generated O2 microbubble mechanism that forms a bubble recoil mechanism, thereby producing a salt-independent motion that overcomes the limitations imposed on the motion of conventional catalytic nanowires by ionic strength. However, it should be noted that this method is not suitable for batch preparation and the velocity of the MNMs it produces is relatively limited.

In section 2.3:

The stress of thin material layers can be exploited to convert the straight band structure into a magnetic helical structure by means of the rolling technique57. For instance, Zhang et al.58 prepared helical MNMs by utilising a conventional thin film deposition method. Initially, the deposited metal layer was converted into straight ribbons by reactive ion etching, followed by chemical vapour deposition to prepare a magnetic head for magnetic control. The formation of the helical structure is attributed to self-rolling action when the straight ribbons are released from the substrate, resulting from the pressure within the material. The parameters of the helical MNMs can be adjusted by varying the deposition conditions, such as the film thickness, the width of the straight ribbons, and the relative orientation of the straight ribbons with respect to the crystalline structure of the metal. In order to further investigate their kinematic properties, Li et al.59 investigated the effect of the magnetic head size of the prepared helical structures on their travelling speed. At lower frequencies, the helical material with a smaller magnetic head moves faster than the helical material with a larger magnetic head due to less viscous resistance in the fluid. However, it was also demonstrated that the spiral substance with the larger head contains more magnetic substance Ni, meaning it will be subjected to a stronger magnetic moment, causing the maximum velocity to increase.

In addition to the utilisation of rotating magnetic fields, helical MNMs prepared by thin film deposition have the capacity to be driven using electro-osmotic forces generated by electric fields60.The principle of electro-osmotic actuation is realised by employing the interface between the surface of the helical structure and the fluid solution. The application of an external field results in the flow of the Stern layer of the helical structure, thereby generating hydrodynamic pressure on the surface of the helical material and propelling the helical structure in the opposite direction.This electro-osmotic actuation of the helical material enhances its travelling speeds and manoeuvrability in comparison to rotating magnetic fields.

  1. However, I miss a part on the experimental motion characterization a discussion on the low-Reynolds number and the usually used mean square displacement (MSD) to characterize Catalytic Micro-/Nanomotors and what are the figures of merits for these systems.

In section 4.2:

Catalytic MNMs are typically reliant on localised chemical reactions to generate propulsive forces, with the reaction process giving rise to an inhomogeneous concentration gradient in the fluid. This, in turn, induces the phenomenon of 'chemical kinetic flow', thereby engendering the autonomous motion of MNMs. However, in low Reynolds number fluid environments, the movement of catalytic MNMs is characterised by a decrease in velocity during the flow process, and the propulsion efficiency and motion pattern are influenced by the fluid viscosity, which may result in slow random wandering or curved paths98,100. In order to enhance the kinematic performance of MNMs, Bai et al. used a rotating magnetic field to improve the kinematic behaviour of helical carbon MNMs with lengths less than 8μm in a low Reynolds number fluid environment110. MNMs exhibit two modes of motion when subjected to a rotating magnetic field: translation and rolling.The translation velocity of MNMs is contingent on the frequency and intensity of the rotating magnetic field, with a maximum velocity of up to 40μm/s being attained.MNMs at low Reynolds numbers typically manifest more intricate motion patterns, particularly with regard to their mean-square displacement (MSD) characteristics111,112. Owing to the prevalence of viscous flow, the trajectories may manifest a strong random and diffusive behaviour, as opposed to exhibiting directional, rapid movement as observed in high Reynolds number environments. In low Reynolds number fluid environments, the MNMs exhibit a considerable component of directional motion for a brief period, and their average velocity is positively correlated with the concentration of fuel molecules. However, over time, the motion reverts to a state of random wandering, accompanied by a significant enhancement in diffusion coefficients 111. The unique nature of low Reynolds number fluid environments necessitates that the design of catalytic micro- and nano-motors must pay particular attention to two aspects. Firstly, how to maximise the local effect of the catalytic reaction and secondly, how to reduce the drag during motion through structural optimisation. The employment of more suitable shapes, surface properties (superhydrophobic or superhydrophilic surfaces), or more efficient catalysts can significantly improve their propulsive performance101,102.

Reviewer 2 Report

Comments and Suggestions for Authors

The authors have put much effort in elaboration of this review. It will help the scientist who is new to the subject to understand the principals of function, of preparation and of use of micro- and nanomotors (MNM). It contains many scheme of mnm that makes it easy to follow the arguments.

In figure 13 they give real values for some properties of MNM. I think, the review would much benefit from supplying more data:

-How high is the speed of buubles or of an object

-How much fuel is consumed for any action

-How is the energy efficiency (in real numbers).

Possibly, no general data may be available, but at least typical data should be provided.

Moreover, the different types of propulsion should be compared with each other, and, if possible to normal engineering propulsions to get an idea of the efficency.

The title in figure 19 contains errors.

Author Response

  1. In figure 13 they give real values for some properties of MNM. I think, the review would much benefit from supplying more data:-How high is the speed of buubles or of an object-How much fuel is consumed for any action -How is the energy efficiency (in real numbers) Possibly, no general data may be available, but at least typical data should be provided.

In section 5.6:

For instance, Mathesh et al. developed an enzyme-driven 2D MNMs utilising a straightforward technique based on soft nanostructures, which exhibited the capacity for autonomous motion at minimal fuel concentrations (0.003% H2O2). When the fuel concentration was increased from 1 mM to 5 mM, the movement velocity of the 2D MNMs increased from 3.5±0.04μm/s to 6.53± 0.68μm/s. The 2D MNMs demonstrated effective positive chemotaxis behaviours and the ability to swim against gravity, owing to the solute buoyancy force. In addition, these catalytic MNMs were able to remove methylene blue dye with up to 85% efficiency137. In order to efficiently assemble MOF NPs into strong MOFs without mechanical stirring, Huang et al. performed large-scale and rapid self-assembly of Fe3O4@NH2-UiO-66 (Fe-UiO) NPs based on Pickering emulsion technique into Fe3O4@NH2-UiO-66 colloidosomes (FeUiOsomes). Through redox action, the MNMs exhibited extreme motility (450 ± 180 μm/s) in a 5 wt% H2O2 solution. The bubble-propelled MNMs were not only able to remove heavy metal ions efficiently (91% for CrVI), but also removed methyl orange up to 94%138.

  1. Moreover, the different types of propulsion should be compared with each other, and, if possible to normal engineering propulsions to get an idea of the efficency.

In section 3.5:

Table2. Comparison of different driving methods for MNMs

Driving method

sports performance

span

biological adaptation

biosafety

limitations

physical

magnetic

Accurate 3D Motion

MNMs are capable of sustained motion under the action of an applied field, with better sustained performance

Good tissue penetration and applicability

Better biocompatibility

Electromagnetic drive with low energy efficiency and limited operating space

light

Start-stop control and directional motion (fast motion)

Poor tissue penetration, only suitable for superficial tissues

UV light is harmful to the body

High-power light sources are required in liquids with limited light focus size and range of motion

ultrasound

fast response time

Good tissue penetration and applicability

comparatively safe

Lack of operational precision

electrical

Convergent movement

Weak bio-penetration, requiring strong electric field strengths

Strong electric fields can affect the human body

Low range of motion

biologically

Relying on the motor properties of individual organisms

Related to biological cell activity

Partially applicable

Biocompatible

Biological cells/bacteria need specific nutrient environments to survive

chemically

High speed of movement

Poor sustained performance due to gradual depletion of chemical fuels

Partially applicable

Common chemical fuels are harmful; enzyme-initiated biocatalytic reactions are biocompatible

Speed of movement depends on fuel concentration; poor persistence; direction of movement uncontrollable

  1. The title in figure 19 contains errors.

I've rearranged the images in the article and fixed spelling mistakes and other types of minor errors in the article. The corrections have been highlighted using a yellow background.

Reviewer 3 Report

Comments and Suggestions for Authors

The manuscript represents a review paper on the current progress in catalytic micro-/nanomotors. It addresses preparation methods, propulsion mechanisms, influencing factors, controlling methods, prospective applications, and biosafety and biocompatibility of these devices.

The manuscript encompasses a broad range of literature from various related scientific disciplines. It is very well written and understandable, even for a reader who is not an expert in micro-/nanomotors.

My one proposition regards the section "MNMs Motion Simulation Analysis." Even though "dynamics and physics governing the motion of MNMs remain insufficiently explored," included equations (6-1) - (6-18) do not fit into the manuscript's overall concept. In my opinion, the manuscript should contain only the conclusions extracted from these equations. In addition, Instructions in the third column of Table 3 incorporate a few non-adequate variable definitions. Please revise these carefully. Also, references to Figures 14D and 14F are permuted in the text.

Other minor typos:

  1. Page 11: Refs 59-65
  2. Page 14: Text: "diffusion coefficient and length"
  3. Fig 12C, D: Please reference these figures in the text.
  4. Subsection 7.1: "Load transport"
  5. Page 37: What "(EC50)" means?
  6. Figure 19 caption: Please correct Ref errors.   

Author Response

  1. My one proposition regards the section "MNMs Motion Simulation Analysis." Even though "dynamics and physics governing the motion of MNMs remain insufficiently explored," included equations (6-1) - (6-18) do not fit into the manuscript's overall concept. In my opinion, the manuscript should contain only the conclusions extracted from these equations.

In the context of simulation studies of catalytic MNMs, the local concentration gradient generated by the catalytic reaction emerges as a pivotal factor influencing motion.The simulation model quantifies the impact of local fluid flow by methodically simulating the consumption of reactants and products generated on the catalyst surface.This simulation encompasses not only the alterations in reactant concentration but also the phenomena of bubble formation, dissolution, thrust, and the diffusion of reaction products during the reaction. In the context of low Reynolds number conditions, the inertial force is rendered negligible, thus rendering the fluid flow predominantly subject to viscous effects. This, in turn, engenders a high degree of randomness and diffusion in the motion of nanowire MNMs within the fluid.The simulation studies indicate that the motion of nanowire motors characteristically manifests as slow random wandering, devoid of stable directional propulsion. In addition to hydrodynamic factors, the geometry, surface smoothness, hydrophilicity or hydrophobicity of the nanowires have a profound effect on their motion performance.For example, elongated nanowires have higher propulsion efficiency compared to spherical or other shaped MNMs because the elongated structure increases the contact area with the fluid and improves the local flow pattern. Further simulation analyses reveal the effect of design parameters (e.g. size, surface modification, catalyst material, etc.) on the performance of nanowire motors. Smaller nanowire MNMs move more slowly in the fluid, but are more flexible and better able to adapt to the microscopic environment, while larger nanowires may have more propulsive force, but their motion is less stable and more susceptible to fluid perturbations.

  1. In addition, Instructions in the third column of Table 3 incorporate a few non-adequate variable definitions. Please revise these carefully.

Table 4. Self-driving MNMs speed expression.

Drive method

Velocity expression equation

Instructions

Self-electrophoretic

Smoluchowsky equation:

ξ: the zeta potential of MNMs;

θ: solution permittivity;

μ: hydrodynamic viscosity;

Nonelectrolyte diffusion

Diffusion swimming model:

k: Boltzmann’s constant;

T: solution temperature;

K:the Gibbs adsorption length;

L:the interaction length of the solute and the motor surface;

C: the solute concentration gradient.

Electrolyte diffusion

Electrophoresis+chemical swimming

: the concentration gradient of the Electrolyte;

: the diffusion rates of cations and anions;

k: the temperature of the T-type solution;

e: the elementary charge;

ɛ: the dielectric constant;

、: the zeta potentials of MNMs and the substrate.

Bubble propulsion

;.

When the bubbles are discharged irregularly, the resultant force Fd of MNMs is 0, and no movement can be produced; when the bubbles are discharged in a certain direction, the resultant force Fd>0 of MNMs moves in a certain direction.

Interfacial tension gradient

where S, R, γ, μ, D, L, and k are the O2 generation rate, radius of MNMs, surface tension of solution, viscosity of solution, diffusion coefficientand length of MNMs, and constant, respectively.

3.Fig 12C, D: Please reference these figures in the text.

The direction of motion of MNMs can be reversed by varying the power of the ultrasound field.The rapid and reversible transition between the aggregated and free-moving states of MNMs in H2O2 fuel is in response to switching on and off the ultrasound state127 (Fig. 12C). The generation of bubbles can be interfered with by the ultrasonic field.Wang et al. achieved reversible control of PEDOT/Ni/Pt microtubule propulsion by varying the applied voltage of an external transducer that generates the ultrasonic field. The authors demonstrated velocity changes of MNMs in a very short time (<0.1 s) and reproducible 'on/off' activations that were faster than those by using other reported methods for stopping the propulsion of MNMs128 (Figure 12D).

I've rearranged the images in the article and fixed spelling mistakes and other types of minor errors in the article. The corrections have been highlighted using a yellow background.